# Lifting Imbalanced Regression with Self-Supervised Learning

## Abstract

A new influential task called imbalanced regression has received a great deal of attention. Different from previous works, this task particularly aims to solve the imbalanced problem in the regression world. The exploration of this task is still at a preliminary stage, so more attempts are needed. In this paper, we first discover that the benefit of Self-supervised Learning (SSL) can help reveal the bias caused by the severe skewness in the training data. Therefore, we work on a seamless marriage of imbalanced regression and SSL. But with this comes the first question of how to measure the similarity and dissimilarity between raw and augmented samples under the regression sense, for which the definition is clear in the classification. To tackle this issue, we provide a formal definition of similarity and dissimilarity in the regression task. In addition, a common practice in SSL to produce augmented samples is to put noise onto the raw samples; the second problem is, it is not guaranteed that the augmented samples are similar to original samples when scaling to a deep network by directly adding random noise to the input. To approach this problem, we specifically limit the volume of noise on the output to get a meaningful augmented sample by back propagation. Experimental results show that our approach achieves state-of-the-art performance.

## 1 Introduction

Regression is a fundamental task in machine learning and statistical analysis, which involves modeling the relationship, between one or more independent and dependent variables. As distinct from the discrete output values in classification, regression makes predictions for continuous values. The issue of imbalanced recognition, arising from the conjunction of imbalance and classification, has been drawing the attention of researchers recently (Shu et al., 2019; Cao et al., 2019; Ren et al., 2020; Jamal et al., 2020; Kang et al., 2020; 2021). By analogy, it is natural to question whether long-tailed regression exists (or one can also call it imbalanced regression, the equivalence of these two names in the paper). Exactly what we expected, a fresh task, namely long-tailed regression, has recently been proposed by Yang et al. (2021). Regardless of the newness of the task, there is no way to neglect its immense significance in terms of application and research. For example, age estimation from facial images; health indicators for various populations in healthcare (Soualhi et al., 2014), these quantities usually exhibit a long-tailed distribution. Going further, in terms of performance, progress in the long-tailed regression research is also far from satisfactory.

More recently, self-supervised learning (SSL) has emerged as a new rage in the research community, owing to its straightforward learning paradigm yet excellent performance (Noroozi & Favaro, 2016; Gidaris et al., 2018; Wu et al., 2018; He et al., 2020). The excellence of the approach has been admired not only on classification with well-balanced datasets, but also surprisingly on long-tailed classification by Kang et al. (2021); Wang et al. (2021); Cui et al. (2021); Samuel & Chechik (2021) who found its benefits. It prompts some thoughts on the potential and practical applications, in particular, applications to SSL for long-tailed regression (Yang et al., 2021). Mindful that the majority of available techniques on SSL are in the context of classification (Oord et al., 2018; Hjelm et al., 2018; Chen et al., 2020a; Henaff, 2020). Due to the difference between classification and regression (no hard boundary existing between outputs in regression task as shown in Figure 1a) (Razi & Athappilly, 2005; Breiman et al., 2017; Liaw et al., 2002; Yang et al., 2021), it is not guaranteed that SSL can help address the imbalanced regression problem well, and it may be difficult to make a straightforward migration from classification to regression task. Upon deeper pondering, there are

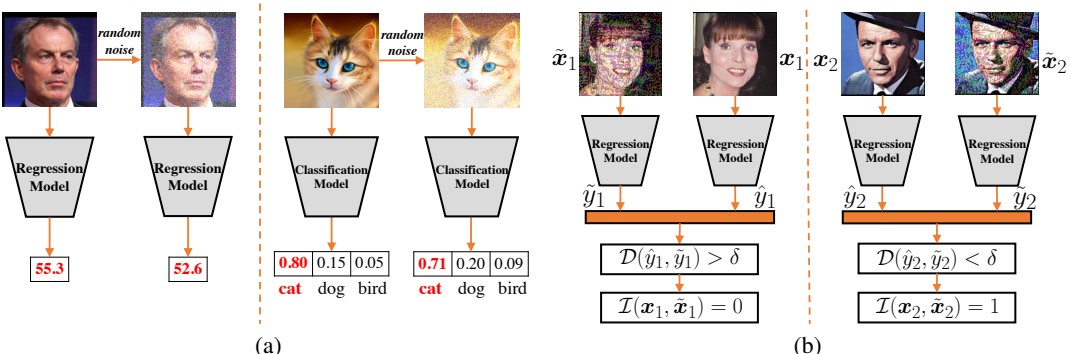

Figure 1: (a) A more visual illustration of the difference between regression and classification with regard to their sensitivity to noise. When noise is imposed, the left image suggests a greater change with augmented data for regression model. The right image remains cat predicted under the same slight perturbation. It reveals the discrepancy between the sensitivity of regression and classification to noise (b) Concepts of similarity and dissimilarity in regression. The right images indicate that after adding noise to the input image, the difference between the age predicted by the augmented image and the original one is less than the threshold $\delta$, so they are considered as similar samples, on the contrary, the difference between the two images on the left is greater than $\delta$, they are therefore recognized as dissimilar samples.

obstacles and problems that we face: *Can SSL really relieve the long-tailed regression problem? If so, how might we solve the imbalanced regression problem by utilizing SSL?*

Figure 1a illustrates the motivation, by adding noise to the image of the cat, it is very unlikely to be misclassified into a class that differs significantly from it, such as a bird, since they are discrete and the categories diverge greatly from each other. In the case of regression, however, it is not at all feasible when a similar migration was made. We imagine that if noise is added to a face image, then the network's estimation of the person's age may be significantly biased. In this regard, it becomes essential to work out and comprehend the degree of noise that induces "similarity and dissimilarity" amongst output values from a regression standpoint. On the foundation of this definition, questions are raised about exploiting noise sensibly and effectively.

With this paper, we try for the first time to propose answers to these questions. And we summarize **our contributions** as follows:

- To the best of our knowledge, we are the first to broaden the concept of similarity and dissimilarity between augmented samples and original samples from classification to regression. And we are also the first to reveal imbalanced regression problem with SSL.

- Having been convinced of the beneficial effect of applying SSL on a simple experiment, we extend it to the neural network-based long-tailed regression task. To ensure that the noise is manageable *w.r.t.* the similarity of the regression, we propose to generate noise on regression values within a predefined threshold.

- We obtain an optimization problem on noise with an efficient approximate solution algorithm based on a first-order Taylor expansion. The augmented input samples are obtained by back-propagation.

- As a practical trial for the proposed method, the results reveal that we attained not only the best results, but also the possibility to mix it freely with other training techniques.

## 2 PRELIMINARY

In this section, we cover the description of imbalanced regression, and then, with reference to the concept of similarity and dissimilarity in classification, we broaden this concept to regression, and provide a new definition of similarity and dissimilarity in regression.

**Notations.** In this paper, lowercase typeface letters (*e.g.*, $x$) represent scalar, lowercase bold typeface letters (*e.g.*, $\boldsymbol{x}$) represent vectors, uppercase typeface letters (*e.g.*, $X$) stand for random variable, uppercase bold typeface letters (*e.g.*, $\boldsymbol{X}$) represent matrix. $\mathrm{diag}(\cdot)$ means diagonalising a vector into a matrix.

## 2.1 PROBLEM STATEMENT

A collection of paired training data $\{(\boldsymbol{x}_i^s, y_i^s)\}_{i=1}^{N_s}$, validation samples $\{(\boldsymbol{x}_i^v, y_i^v)\}_{i=1}^{N_v}$ and testing samples $\{(\boldsymbol{x}_i^t, y_i^t)\}_{i=1}^{N_t}$ are presumed to be available, where $\boldsymbol{x}_i^o \in \mathbb{R}^d, o \in \{s, v, t\}$, and $y_i^o$ are a continuous variables. We suppose that $y_i^o$ is well-bounded and notate the maximum and minimum values of $y_i^o$ as $y_{max}^o$ and $y_{min}^o$ accordingly. Further, we are able to partition $y_i^o$ into $B$ bins $[y_0^o, y_1^o), [y_1^o, y_2^o), \ldots, [y_{B-1}^o, y_B^o)$ with equal interval $\Delta y = \frac{y_{max}^o - y_{min}^o}{B}$. In so doing we could put every $y_i^o$ in the dataset into the corresponding bin fulfilling the condition $y_b^o < y_i^o \le y_{b+1}^o$, where $b \in \{0, 1, \ldots, B-1\}$. With three splits (training, validation, testing), if we count the number of data points that fall into each bin and form these into a vector $\boldsymbol{m}^o = (m_1^o, \ldots, m_B^o)$. The *imbalanced character* of the normal regressions lies in the existence of $m_{b_i}^s, m_{b_j}^s \in \boldsymbol{m}^s$, allowing $m_{b_i}^s \gg m_{b_j}^s$ to hold. It is the opposite of this that the values of the elements of the two vectors $\boldsymbol{m}^v$ and $\boldsymbol{m}^t$, are approximately equal. A desire for imbalanced regression enables model $f(\boldsymbol{x}_i^o; \boldsymbol{\theta})$ produced on an imbalanced training set to be potentially generalized to a balanced validation and testing set to the maximum extent possible. Typically, $f(\boldsymbol{x}_i^o; \boldsymbol{\theta})$ may be a neural network with parameter $\boldsymbol{\theta}$ because of its impressive fitting ability. In what follows, for the sake of convenience, we will *drop the superscript o*.

In the introduction section, we elaborate on the predicament that the lack of a clear definition of the concepts of similarity and dissimilarity in the regression, as regression involves continuous targets, where hard boundaries between classes do not exist. As a solution to this problem, we formally propose the following definition:

**Definition 1** *There exists some threshold $\delta$, regarding two data samples $(\boldsymbol{x}_i, \hat{y}_i), (\tilde{\boldsymbol{x}}_i, \tilde{y}_i)$, $\tilde{\boldsymbol{x}}_i$ is obtained by adding some noise $\boldsymbol{r}$ to $\boldsymbol{x}_i$, and $\tilde{y}_i$ and $\hat{y}_i$ is inferred by the model $f(;\boldsymbol{\theta})$ from $\tilde{\boldsymbol{x}}_i$ and $\boldsymbol{x}_i$ respectively. We remark that $(\boldsymbol{x}_i, \hat{y}_i), (\tilde{\boldsymbol{x}}_i, \tilde{y}_i)$ are similar samples of the model $f(\boldsymbol{x}_i; \boldsymbol{\theta})$ (no need for well-trained model) if the distance between the two samples is less than $\delta$ under some distance function $\mathcal{D}(\cdot, \cdot)$, such as $l_1$ or $l_2$ distance. Mathematically, it can be phrased in this following:*

$$\mathcal{I}(\boldsymbol{x}_i, \tilde{\boldsymbol{x}}_i) = \begin{cases} 1, & if \quad \mathcal{D}(\hat{y}_i, \tilde{y}_i) < \delta \\ 0, & if \quad \mathcal{D}(\hat{y}_i, \tilde{y}_i) \ge \delta, \end{cases} \tag{1}$$

where $\mathcal{I}(\boldsymbol{x}_i, \tilde{\boldsymbol{x}}_i)$ indicates the trueness of the condition whether two samples $\boldsymbol{x}_i$ and $\tilde{\boldsymbol{x}}_i$ are similar samples. The parameter $\delta$ here is manually specified and task-dependent. We display a virtual example of this definition as illustrated in Figure 1b. We select images from AgeDB-DIR and attach noises to them, afterward we estimate the ages of the original images and their corresponding noisy images with the network. Finally, we identify a pair of original image and noisy image as not similar if their difference exceeds the threshold $\delta$, and vice versa.

## 3 MOTIVATION

Herewith we illustrate, for simple shallow neural networks, that SSL facilitates long-tailed regression in a positive way. Aside from this, we would characterize the difficulties involved in extending our approach to deep neural networks, *i.e.* for controlling the level of noise to guarantee the creation of similar samples under the sense of regression. We first describe the data generation, together with a description of the specific experimental settings followed by an analysis. The network structure and implementation details are available in the appendix.

**Data generation.** Consider two components of data following two different Gaussian distribution, in particular, $\boldsymbol{x}_1 \sim \mathcal{N}(\mu_1 \cdot \mathbf{1}_d, \mathrm{diag}(\sigma_1 \cdot \mathbf{1}_d))$, $\boldsymbol{x}_2 \sim \mathcal{N}(\mu_2 \cdot \mathbf{1}_d, \mathrm{diag}(\sigma_2 \cdot \mathbf{1}_d))$. $\mathbf{1}_d$ denotes a vector with all 1's. The values of $\mu_1(\sigma_1)$ and $\mu_2(\sigma_2)$ are quite different. With linear mapping to $\boldsymbol{x}_1$ and $\boldsymbol{x}_2$ with parameter $\boldsymbol{w} \in \mathbb{R}^d$, it yields the output value $y = \boldsymbol{w}^\mathrm{T}\boldsymbol{x}$. As a result, we get two collections of training data $\{\boldsymbol{x}_{1,i}^s, y_{1,i}^s\}_{i=1}^{N_1^s}$, $\{\boldsymbol{x}_{2,j}^s, y_{2,j}^s\}_{j=1}^{N_2^s}$. Suppose $N_1^s > N_2^s$, and we measure

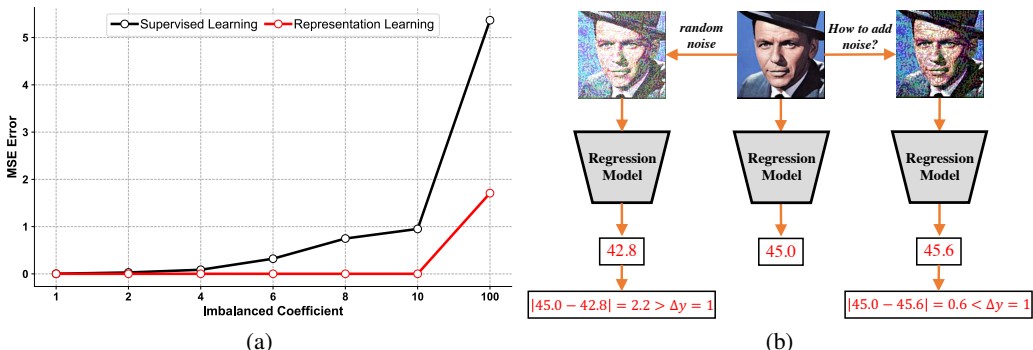

Figure 2: (a) Mean Square Error with different imbalanced coefficients. Very clearly it is observed that under supervised learning, MSE grows as the imbalanced coefficient increases, contrary to the SSL case where MSE is considerably smaller with the same imbalanced coefficient. (b) A demonstration of the impact of noise generation on the regression values, when adding random noise to the original image, the difference between the output value of noisy image versus such a component of the original image will most likely exceed the threshold.

imbalance here in terms of $R = \frac{N_1^s}{N_2^s}$. We pick up the value of $R$ in $\{1, 2, 4, 6, 8, 10, 100\}$. Despite the training samples are imbalanced, the testing set $\{\boldsymbol{x}_{1,i}^t, y_{1,i}^t\}_{i=1}^{N_1^t}, \{\boldsymbol{x}_{2,j}^t, y_{2,j}^t\}_{j=1}^{N_2^t}$ we generate is balanced. Additionally, we produce similar samples by controlling the noise so that the condition $\min(y_{1,i}, y_{2,j}) \gg \delta, \forall i \in \{1, \dots, N_1\}, j \in \{1, \dots, N_2\}$ is satisfied.

**Experimental settings.** We compare two training methods here: (i) *Representation Learning*: It is trained in the first $T_1$ epochs using a self-supervised approach, and the resulting network is treated as an initialized network, and in the second $T_2$ epochs it is trained in a standard supervised learning manner.(ii) *Supervised Learning*: it is entirely trained using supervised learning in the $T_1 + T_2$ epochs. As per the normal process, the model is trained on the training set and tested on the testing set. To evaluate we use Mean Square Error (MSE) as a criterion. Here we use the vanilla mean squared loss function.

**Experimental findings.** We plot the variation curve of MSE with the imbalance coefficient on the testing set for the two training methods. As illustrated in Figure 2a, what has been clearly observed is that the MSE for both methods grows with an increase in the imbalance coefficient. More importantly, it is found that SSL could alleviate this problem to some degree. The experimental findings, thus, lead us to mitigate the imbalance regression problem with SSL.

**Remark** It is true that the above experiments concede that SSL can be beneficial to long-tailed regression, albeit with some restrictions that remain when we adapt it to neural networks. Well, yes, we have imposed settings for noise, namely, the noise added is controllable in terms of its impact on the regression values. Unfortunately, scaling the technique beyond shallow to deep networks, there is a difficulty in keeping the difference between the output values of similar samples under regression within a threshold, a concern that arises in the case of a sample augmented with random noise, where the difference between the output values of the noisy sample and the original sample is more than the threshold, as shown in Figure 2b, *hence the challenge stems from finding valid noise to ensure the generation of similar samples*.

## 4 METHOD

Just as we showed before, it is not feasible to merely port the above method to regression tasks. Instead, it will backfire if at all we do so. Hence, it is essential to control the noise $\boldsymbol{r}$ to limit the gap between $\hat{y}_i$ and $\tilde{y}_i$ in the length of a bin $\Delta y$. We search for the most anisotropic noise causing the generated samples to be similar or dissimilar to the original samples, which means that we have to impose constraints on the output values. Rather than adding noise to the input, we blaze new a trail, that is, we apply noise to the output value. Mathematically, it can be expressed by the following

formula:

$$\arg\min_{\boldsymbol{\theta}} \arg\max_{\boldsymbol{r}_i} \mathcal{D}(\hat{y}_i, \tilde{y}_i; \boldsymbol{\theta}), \tag{2}$$

$$s.t. \quad ||\boldsymbol{r}_i|| < \epsilon, \tag{3}$$

$$\text{where} \quad \hat{y}_i = f(\boldsymbol{x}_i; \boldsymbol{\theta}) \quad \text{and} \quad \tilde{y}_i = \hat{y}_i + z_i \quad \text{and} \quad \tilde{y}_i = f(\boldsymbol{x}_i + \boldsymbol{r}_i; \boldsymbol{\theta}) \tag{4}$$

where $\mathcal{D}$ denotes the distance loss function, and $\tilde{y}_i$ is the noisy output *w.r.t.* noises $z_i$ and $\boldsymbol{r}_i$.

Here $z_i$ is randomly sampled from $[-\Delta y, \Delta y]$. This allows us to assure that $\tilde{y}_i$ and $\hat{y}_i$ are similar, while simultaneously enhancing the local smoothness of the output. Normally it is not possible to obtain $\boldsymbol{r}$ to find a closed-form solution. Since neural networks are highly dimensional and nonlinear, the analytical solutions can not be derived for $\boldsymbol{r}$ . For the purpose of facilitating analysis and simultaneously providing efficient training of the network, we approximate $\mathcal{D}(\hat{y}_i, \tilde{y}_i; \boldsymbol{\theta})$ by the first-order Taylor expansion:

$$\mathcal{D}(\hat{y}_i, \tilde{y}_i; \boldsymbol{\theta}) \approx \nabla_{\boldsymbol{x}_i} \mathcal{D}(\hat{y}_i, \tilde{y}_i; \boldsymbol{\theta})^{\mathrm{T}} (\tilde{\boldsymbol{x}}_i - \boldsymbol{x}_i) + a = \nabla_{\boldsymbol{x}_i} \mathcal{D}(\hat{y}_i, \tilde{y}_i; \boldsymbol{\theta})^{\mathrm{T}} \boldsymbol{r}_i + a \tag{5}$$

$$s.t. \quad ||\boldsymbol{r}_i|| < \epsilon \tag{6}$$

Where $a = \mathcal{D}(\hat{y}_i, \tilde{y}_i; \boldsymbol{\theta})|_{\boldsymbol{x}=\boldsymbol{x}_i}$ is a constant term around $\boldsymbol{x}_i$. Based on this, the close-form solution of $\boldsymbol{r}_i$ can be calculated as

$$\boldsymbol{r}_i := \arg\max_{\boldsymbol{r}_i:||\boldsymbol{r}_i||\leq\epsilon} \nabla_{\boldsymbol{x}_i} \mathcal{D}(\hat{y}_i, \tilde{y}_i; \boldsymbol{\theta})^{\mathrm{T}} \boldsymbol{r}_i + a \tag{7}$$

$$= \frac{\nabla_{\boldsymbol{x}_i} \mathcal{D}(\hat{y}_i, \tilde{y}_i; \boldsymbol{\theta})}{||\nabla_{\boldsymbol{x}_i} \mathcal{D}(\hat{y}_i, \tilde{y}_i; \boldsymbol{\theta})||} ||\boldsymbol{r}_i||_{\max} \tag{8}$$

$$s.t. \quad ||\boldsymbol{r}_i|| \leq \epsilon \tag{9}$$

Therefore, it is obviously that $\boldsymbol{r}_i = \epsilon \frac{\nabla_{\boldsymbol{x}_i} \mathcal{D}(\hat{y}_i, \tilde{y}_i; \boldsymbol{\theta})}{||\nabla_{\boldsymbol{x}_i} \mathcal{D}(\hat{y}_i, \tilde{y}_i; \boldsymbol{\theta})||}$. Furthermore, as Eq. 4 presented, $\tilde{y}_i = \hat{y}_i + z_i$, so $\boldsymbol{r}_i$ can be further denoted as

$$\boldsymbol{r}_i = \epsilon \frac{\nabla_{\boldsymbol{x}_i} \mathcal{D}(\hat{y}_i, \hat{y}_i + z_i; \boldsymbol{\theta})}{||\nabla_{\boldsymbol{x}_i} \mathcal{D}(\hat{y}_i, \hat{y}_i + z_i; \boldsymbol{\theta})||} \tag{10}$$

---

**Algorithm 1:** Self-Supervised Imbalanced Regression (SSIR)

---

**Input:** Mini-batch $\boldsymbol{B}$ with $M$ samples, model $f(\cdot; \boldsymbol{\theta})$ with its parameters $\boldsymbol{\theta}$, bin length $\Delta y$, limitation coefficient of noise $\epsilon$, divergence distance function $\mathcal{D}$, values range of input samples $(v_{min}, v_{max})$, function $clip(\cdot, v_{max}, v_{min})$ to clip values that out of range of $(v_{min}, v_{max})$

**Result:** Updated parameters of the model $\hat{\boldsymbol{\theta}}$.

1 **for** $\boldsymbol{x}_i$ *in* $\boldsymbol{B}$ **do**
2      Initial random noise $z_i$, *s.t.* $|z_i| < \Delta y$;
3      $\hat{y}_i = f(\boldsymbol{x}_i; \boldsymbol{\theta})$, $\tilde{y}_i = \hat{y}_i + z_i$;
4      Calculate $\boldsymbol{r}_i$ by taking the gradient of $\mathcal{D}(\hat{y}_i, \tilde{y}_i; \boldsymbol{\theta})$ *w.r.t.* $\boldsymbol{x}_i$ :
5          $\boldsymbol{g}_i = \nabla_{\boldsymbol{x}_i} \mathcal{D}(\hat{y}_i, \tilde{y}_i; \boldsymbol{\theta}) = \nabla_{\boldsymbol{x}_i} \mathcal{D}(f(\boldsymbol{x}_i; \boldsymbol{\theta}), f(\boldsymbol{x}_i; \boldsymbol{\theta}) + z_i)$
6          $\boldsymbol{r}_i \leftarrow \epsilon \times \boldsymbol{g}_i / ||\boldsymbol{g}_i||_2$;
7      Update $\tilde{y}_i$ by taking $\boldsymbol{x}_i + \boldsymbol{r}_i$ as the input of $f$:
8          $\tilde{\boldsymbol{x}}_i = clip(\boldsymbol{x}_i + \boldsymbol{r}_i, v_{max}, v_{min})$;
9          $\tilde{y}_i = f(\tilde{\boldsymbol{x}}_i; \boldsymbol{\theta})$;
10     Update $\boldsymbol{r}_i$ by by taking the gradient of $\mathcal{D}(\hat{y}_i, \tilde{y}_i; \boldsymbol{\theta})$*w.r.t.* $\boldsymbol{x}_i$ :
11         $\boldsymbol{g}_i = \nabla_{\boldsymbol{x}_i} \mathcal{D}(\hat{y}_i, \tilde{y}_i; \boldsymbol{\theta})$
12         $\boldsymbol{r}_i \leftarrow \epsilon \times \boldsymbol{g}_i / ||\boldsymbol{g}_i||_2$;
13     $\tilde{\boldsymbol{x}}_i = clip(\boldsymbol{x}_i + \boldsymbol{r}_i, v_{max}, v_{min})$
14     $\tilde{y}_i = f(\tilde{\boldsymbol{x}}_i; \boldsymbol{\theta})$
15 **end**

16 **return** $\hat{\boldsymbol{\theta}} \leftarrow \boldsymbol{\theta} - \nabla_{\boldsymbol{\theta}}(\frac{1}{M} \sum_{i=1}^{M} \mathcal{D}(\hat{y}_i, \tilde{y}_i, \boldsymbol{\theta}))$

---

$\tilde{x} = x + r$ obtained in this way, however, may exceed the boundary value of the feature (For example, the range of image pixels is $[0, 255]$). To cope with such an accident, we constrain the feature values to this range by clipping their values. On the other hand, considering that a single back-propagation optimization may not be an accurate approximation, we therefore performed a two-step optimization of $r$. In our experiments, one-step optimisation approach does not adequately yield optimal performance, and more iterations will introduce more computational cost to training without significant performance gain. Up to now, our proposed *Self-Supervised Imbalanced Regression* (SSIR) algorithm is completely introduced. In order to show the SSIR more clearly. We give the algorithm workflow in Algorithm 1.

Our training strategy includes normal regression training and self-supervised regression. For the regression training, we directly apply supervised training to minimize the loss function between the outputs of the model $\hat{y}$ and the corresponding ground truth $y$. Besides, an additional coefficient $\lambda$ is introduced to the final loss. Our overall training function is shown below:

$$\mathcal{L}_{joint} = \mathcal{L}_R + \lambda\mathcal{L}_{SSIR} \tag{11}$$

$$= \frac{1}{N} \sum_{i=1}^{N} [\mathcal{D}(\hat{y}_i, y_i; \boldsymbol{\theta}) + \lambda\mathcal{D}(\hat{y}_i, \tilde{y}_i; \boldsymbol{\theta})] \tag{12}$$

$$= \frac{1}{N} \sum_{i=1}^{N} [\mathcal{D}(f(\boldsymbol{x}_i; \boldsymbol{\theta}), y_i) + \lambda\mathcal{D}(f(\boldsymbol{x}_i; \boldsymbol{\theta}), f(\boldsymbol{x}_i + \boldsymbol{r}_i; \boldsymbol{\theta}))] \tag{13}$$

## 5 EXPERIMENTS

In this section, we report the results on three datasets to demonstrate the superiority of our proposed methods, when compared with state-of-the-art baselines. Specifically, we compare our method with **VANILLA**, **SMOTER** (Torgo et al., 2013), **SMOGN** (Branco et al., 2017), **RRT** (Yang et al., 2021), **INV**, **SQINV**, **FOCAL-R** (Yang et al., 2021), **LDS** (Yang et al., 2021), **FDS** (Yang et al., 2021) and some of their combinations on three imbalanced regression datasets including **IMDN-WIKI-DIR**, **AgeDB-DIR** and **NYUD2-DIR**. Note that, in all the experiments, the **LDS** is combined with re-weighting methods *i.e.* **SQINV** or **INV**. In otherwise, our proposed SSIR can be combined with various methods in training process that demonstrates the flexibility of SSIR. The evaluation metrics used on **IMDB-WIKI-DIR** and **AgeDB-DIR** datasets are **MAE** and **GM** (Yang et al., 2021), besides, **RMSE** and $\boldsymbol{\delta_1}$ are also employed on **NYUD2-DIR** dataset. For the details of datasets, baselines, evaluation metrics and implementation details, we put them in the appendix.

Table 1: Comparison on the MAE and GM with several competed baselines on IMDB-WIKI-DIR dataset. The underlined part denotes the second-best result, and the bold part denotes the best result.

| Method | MAE | | | | GM | | | |
| --- | --- | --- | --- | --- | --- | --- | --- | --- |
| | All | Many | Med. | Few | All | Many | Med. | Few |
| VANILLA (Yang et al., 2021) | 8.06 | 7.23 | 15.12 | 26.33 | 4.57 | 4.17 | 10.59 | 20.46 |
| SMOTER (Torgo et al., 2013) | 8.14 | 7.42 | 14.15 | 25.28 | 4.64 | 4.30 | 9.05 | 19.46 |
| SMOGN (Branco et al., 2017) | 8.03 | 7.30 | 14.02 | 25.93 | 4.63 | 4.30 | 8.74 | 20.12 |
| SMOGN + LDS + FDS (Yang et al., 2021) | 7.97 | 7.38 | 13.22 | 22.95 | 4.59 | 4.39 | 7.84 | 14.94 |
| RRT (Yang et al., 2021) | 7.81 | 7.07 | 14.06 | 25.13 | 4.35 | 4.03 | 8.19 | 16.96 |
| RRT + LDS + FDS (Yang et al., 2021) | 7.65 | 7.06 | 12.41 | 23.51 | 4.31 | 4.07 | 7.17 | 15.44 |
| INV (Yang et al., 2021) | 8.17 | 7.64 | 12.46 | 22.83 | 4.70 | 4.51 | 6.94 | 13.78 |
| SQINV (Yang et al., 2021) | 7.87 | 7.24 | 12.44 | 22.76 | 4.47 | 4.22 | 7.25 | 15.10 |
| SQINV + LDS + FDS (Yang et al., 2021) | 7.78 | 7.20 | 12.61 | **22.19** | 4.37 | 4.12 | 7.39 | **12.61** |
| **SQINV + SSIR** | **7.63** | **7.06** | **12.17** | 23.10 | **4.26** | 4.02 | **6.82** | 14.93 |
| FOCAL-R (Yang et al., 2021) | 7.97 | 7.12 | 15.14 | 26.96 | 4.49 | 4.10 | 10.37 | 21.20 |
| FOCAL-R + LDS + FDS (Yang et al., 2021) | 7.88 | 7.10 | 14.08 | 25.75 | 4.47 | 4.11 | 9.32 | 18.67 |
| **FOCAL-R + SSIR** | 7.88 | 7.08 | 14.75 | 25.16 | 4.39 | **4.01** | 10.20 | 18.04 |

## 5.1 Main Results

In the experiments on IMDB-WIKI-DIR dataset, we combine our proposed SSIR with SQINV and FOCAL-R respectively. As shown in Table 1, our method achieves the best performance with **7.63** and **4.26** on MAE and GM accordingly. Compared with the second-best method RRT + LDS + FDS, there has improvements of 0.02 and 0.05 in MAE and GM with our approach. Besides, in comparison with other FOCAL-R-based methods, greater performance can be achieved with the combination of FOCAL-R and SSIR with a better GM score of **4.39** and the MAE score of **7.88**. For re-weighting-based methods, the combination of SQINV and SSIR outperforms the second-best re-weighting-based method SQINV + LDS + FDS and improves **0.15** and **0.11** on MAE and GM respectively.

Table 2: Comparison on the MAE and GM with several competed baselines on AgeDB-DIR dataset. The underlined part denotes the second-best result, and the bold part denotes the best result.

| Method | MAE | | | | GM | | | |
|---|---|---|---|---|---|---|---|---|
| | All | Many | Med. | Few | All | Many | Med. | Few |
| VANILLA (Yang et al., 2021) | 7.77 | 6.62 | 9.55 | 13.67 | 5.05 | 4.23 | 7.01 | 10.75 |
| SMOTER (Torgo et al., 2013) | 8.16 | 7.39 | 8.65 | 12.28 | 5.21 | 4.65 | 5.69 | 8.49 |
| SMOGN (Branco et al., 2017) | 8.26 | 7.64 | 9.01 | 12.09 | 5.36 | 4.90 | 6.19 | 8.44 |
| SMOGN + LDS + FDS (Yang et al., 2021) | 7.90 | 7.32 | 8.51 | 11.19 | 4.98 | 4.64 | 5.41 | 7.35 |
| RRT (Yang et al., 2021) | 7.74 | 6.98 | 8.79 | 11.99 | 5.00 | 4.50 | 5.88 | 8.63 |
| RRT + LDS + FDS (Yang et al., 2021) | 7.66 | 6.99 | 8.60 | 11.32 | 4.80 | 4.42 | 5.53 | 6.99 |
| INV (Yang et al., 2021) | 7.97 | 7.31 | 8.81 | 11.62 | 5.05 | 4.64 | 5.75 | 8.20 |
| SQINV (Yang et al., 2021) | 7.81 | 7.16 | 8.80 | 11.20 | 4.99 | 4.57 | 5.73 | 7.77 |
| SQINV + LDS + FDS (Yang et al., 2021) | 7.55 | 7.01 | **8.24** | **10.79** | 4.72 | 4.36 | 5.45 | **6.79** |
| **SQINV + SSIR** | 7.49 | 6.81 | 8.54 | 11.04 | **4.61** | 4.28 | **5.12** | 6.92 |
| FOCAL-R (Yang et al., 2021) | 7.64 | 6.68 | 9.22 | 13.00 | 4.90 | 4.26 | 6.39 | 9.52 |
| FOCAL-R + LDS + FDS (Yang et al., 2021) | 7.47 | 6.69 | 8.30 | 12.55 | 4.71 | 4.25 | 5.36 | 8.59 |
| **FOCAL-R + SSIR** | **7.35** | **6.53** | 8.51 | 11.81 | 4.62 | **4.09** | 5.64 | 8.46 |

In the experiments on AgeDB-DIR, with the same experimental settings on IMDB-WIKI-DIR, we combine our proposed SSIR with SQINV and FOCAL-R respectively. Table 2 illustrates the main results of our method and baselines on AgeDB-DIR dataset. Clearly our method achieves the best performance on both MAE and GM with **7.35** and **4.61** respectively. As the combination of the re-weighting method and SSIR, SQINV + SSIR performs better than the other four re-weighting-based methods with 0.06 and **0.11** improvements on MAE and GM separately, compared to the second-best re-weighting-based method SQINV + LDS + FDS. For FOCAL-R-based methods, the association with SSIR can further decrease the testing error than FOCAL-R and FOCAL-R + LDS + FDS.

For NYUD2-DIR dataset, we combine SSIR with VANILLA and re-weighting method INV respectively. As presented in Table 3, our method INV + SSIR achieves the best performance among other methods. Specifically, INV + SSIR outperforms the second-best method INV + LDS + FDS on RMSE with 0.047, and our method achieves the best performance on $\delta_1$ with 0.075. Compared with VANILLA, the combination with SSIR can improve the performance significantly with 0.07 on RMSE and 0.007 on $\delta_1$.

Table 3: Comparison on the RMSE and $\delta_1$ with several competed baselines on NYUD2-DIR dataset. The underlined part denotes the second-best result, and the bold part denotes the best result.

| Method | RMSE | | | | $\delta_1$ | | | |
|---|---|---|---|---|---|---|---|---|
| | All | Many | Med. | Few | All | Many | Med. | Few |
| VANILLA (Yang et al., 2021) | 1.477 | 0.591 | 0.952 | 2.123 | 0.677 | **0.777** | 0.693 | 0.570 |
| **VANILLA + SSIR** | 1.407 | **0.580** | 0.883 | 2.022 | 0.684 | 0.772 | 0.704 | 0.590 |
| INV + LDS + FDS (Yang et al., 2021) | 1.338 | 0.670 | **0.851** | 1.880 | 0.705 | 0.730 | **0.764** | 0.655 |
| **INV + SSIR** | **1.291** | 0.766 | 0.894 | **1.756** | **0.705** | 0.669 | 0.763 | **0.718** |

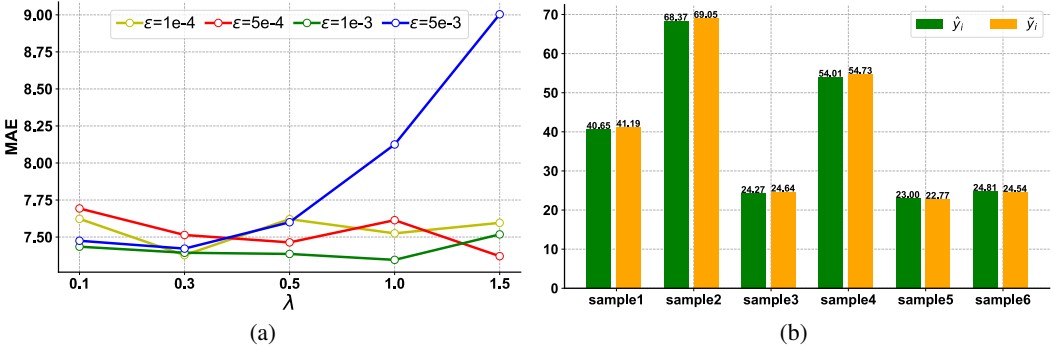

(a)                                                (b)

Figure 3: (a) Parameter study of **FOCAL-R + SSIR** on AgeDB-DIR dataset, for a fixed value of $\lambda$, we vary the value of $\epsilon$. (b) Original outputs and noisy outputs of samples from AgeDB-DIR dataset.

## 5.2 PARAMETER AND CASE STUDY

**Parameter Study.** Figure 3a illustrates the results of parameter study of **FOCAL-R + SSIR** on AgeDB-DIR dataset. As it shows, each line denotes the results of different $\lambda$ when fixing $\epsilon$, and we conduct such experiments with the $\epsilon$ of 5e-3, 1e-3, 5e-4 and 1e-4. The overall results show that our method achieves the best performance with $\epsilon = 1e-3$ and $\lambda = 1.0$.

**Case Study.** We also conduct case studies to visualize some results of selected examples. As Figure 3b shows, the gaps between the outputs and their associated noisy outputs of all the chosen 6 examples do not exceed a bin length, which means our proposed method can successfully control the noise to limit the noisy outputs within a reasonable range.

**Visualization on Noisy Samples.** Moreover, we also show three examples and their corresponding noisy examples with $\epsilon$ of 0.05, 0.1, 0.3, 0.5 and 1.0 respectively in Figure 4. Obviously, with the $\epsilon$ increasing, the noisy examples become unclearer that means the noise level is positively correlated with $\epsilon$.

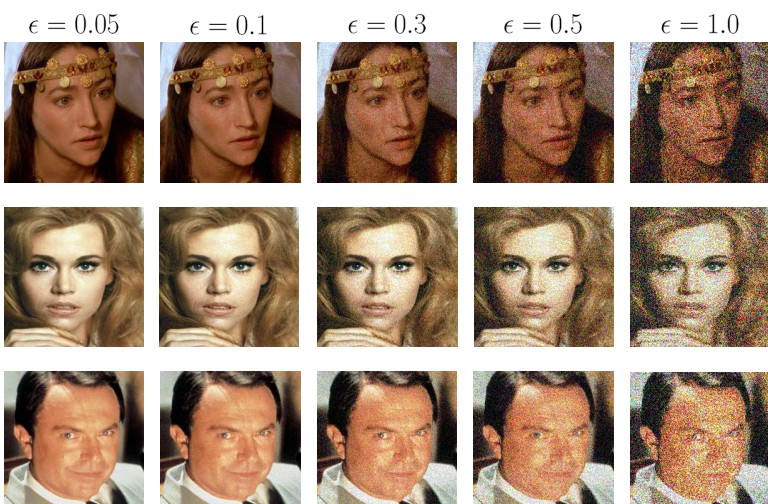

Figure 4: Three groups of noisy samples generated from three samples chosen from AgeDB-DIR dataset with different $\epsilon$.

## 6    RELATED WORK

Presented here are the literature on imbalance problems and SSL. Concerning imbalance problems, the introduction focuses on classification and regression. For SSL, we review some of the advances and applications.

### 6.1    IMBALANCED PROBLEM

From the object of imbalanced problems, it can be divided into two categories: imbalanced classification and imbalanced regression.

**Imbalanced classification.** In the imbalanced classification task, existing methods mainly adopt data-based or model-based solutions (Yang et al., 2021). The core idea of data-based methods is reducing the gap of sampling numbers between many-shot classes and few-shot classes (Chawla et al., 2002; He et al., 2008). For the model-based methods, several works have been proposed to tackle such challenges based on lots of aspects. Jamal et al. (2020) proposed to solve long-tailed classification through domain adaptation perspective. Kang et al. (2020) used a two-stage learning strategy to learn representations for the classification layer using a long-tailed training set. Kang et al. (2021) explored the impact of balanced feature spaces when dealing with imbalanced data, and proposed a SSL-based method to better learn the representation for imbalanced classification. Besides, SSL on unbalanced classification was similarly explored in Wang et al. (2021); Cui et al. (2021); Samuel & Chechik (2021). Furthermore, there also has some methods addressing the problem by well-designed loss functions (Zhang et al., 2017; Cao et al., 2019; Cui et al., 2019; Ren et al., 2020) and other learning paradigms, *i.e.* meta learning (Shu et al., 2019; Wang et al., 2020).

**Imbalanced regression.** Imbalanced problem has also been explored in regression problems. Conventional works mainly applied data-based methods to solve the imbalanced problem in regression tasks. One of the first works to address this problem, is published by Torgo et al. (2013) using SMOTE (Chawla et al., 2002). Branco et al. (2017) proposed a Gaussian noise-based synthetic case generation method for imbalanced regression. After that, they further proposed a bagging-based method to aggregate various data pre-processing strategies (Branco et al., 2018) for the same task. Recently, Yang et al. (2021) further delved into such problem by analyzing the error distribution and the statistic distribution of features, both *w.r.t.* labels, and proposed two distribution smoothing methods for labels and features. And they also proposed the Focal-R, a new loss function for imbalanced regression, motivated by Focal loss (Lin et al., 2017).

### 6.2    SELF-SUPERVISED LEARNING

Recently, SSL has received extensive attention in the field of deep learning. Different from conventional unsupervised learning and supervised learning, SSL can use the semantic or contexture supervision information to learn robust and well-generalization representations without manual annotations, include contrastive learning-based methods (Chen et al., 2020a;b;c; He et al., 2020) and some other pretext tasks-based methods, such as image inpainting (Pathak et al., 2016; Jenni & Favaro, 2018), context prediction (Doersch et al., 2015), rotation prediction (Gidaris et al., 2018) and so on. Whereas we tackle the optimization problem in a very similar way, the difference concerns the fact that we apply noise to the output directly rather than to the input, as well as the fact that the problem we analyze varies from semi-supervised classification problems to imbalanced regression problems.

## 7    CONCLUSION

In this paper, we explore the Self-Supervised Learning (SSL) in imbalanced regression scenario. Specifically, we propose *Self-Supervised Imbalanced Regression* (SSIR) to better address the imbalanced problem in regression tasks. In SSIR, instead of adding noise to the inputs directly to generate the noisy samples, we add noise, which is limited by the bin size, to the output of the model to avoid the excessive dissimilarity between samples and noisy samples, followed by back propagation steps to obtain the meaningful noisy samples. This work provides a new paradigm to tackle this challenge. Further, extensive experiments have shown the superiority of our proposed SSIR over existing state-of-the-art baselines.

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

# 8 APPENDIX

In the appendix, we first offer two comparative experiments. Then we give the implementation details in the motivation, as well as the details of the datasets and baseline methods included in the experiments. Besides, we also present more detailed experimental results of parameter studies, and loss curves within different settings. Finally, We investigate the beneficial impact of our approach on feature imbalance.

## 8.1 COMPARISON WITH RANDOM NOISE METHOD

With the addition of random noise, the combined SSIR with FOCAL-R and SQINV approaches can achieve the performance on MAE with **7.57** and **7.71** respectively on AgeDB-DIR dataset. In contrast, the combined SSIR with FOCAL-R and SQINV approaches with noise generation proposed in our paper can achieve MAE performance with **7.35** and **7.49** respectively. The experimental results demonstrate the superiority of our noise generation approach.

## 8.2 COMPARISON WITH ONE-STEP OPTIMIZATION METHOD

To quantitatively compare the benefits of two-step approximation against one-step approximation, we conducted a comparison experiment on the AgeDB-DIR dataset. The setup of the comparison experiment is following: we perform a one-step optimization of $r$ and provide results combined with SQINV; except for this, we keep the other hyperparameters consistent with our experiments. When performing one-step optimization, the combined SSIR with SQINV approach can achieve the performance on MAE with **7.62**. In contrast, the combined SSIR with SQINV approach with a two-step approximation can achieve MAE performance with **7.49**. The experimental results demonstrate the superiority of two-step approximation. However, we have found in our experiments that there is no significant increase in performance but an increase in computational overhead when the number of optimization steps is greater than or equal to 3. Therefore, in order to balance accuracy and efficiency, we have used two-step optimization in our paper.

## 8.3 EXPERIMENTS SETTINGS

**Datasets** We evaluate our proposed method on three large-scale deep imbalanced regression datasets, the tasks of which are age estimation and depth estimation. For all the datasets used in our experiments, we follow the settings in (Yang et al., 2021) and segment the target bins into three subsets: many-shot which contains more than 100 training images, medium-shot in which the number of training samples in each bin is in the range of 20 to 100, and few-shot that contains less than 20 training samples in each bin. The details of the datasets are as following:

- **IMDB-WIKI-DIR:** IMDB-WIKI-DIR is an imbalanced age estimation dataset. This dataset is created by Yang et al. (2021) based on the IMDB-WIKI dataset (Rothe et al., 2018). The original IMDB-WIKI contains 523.0K face images with associated ages. In IMDB-WIKI-DIR, the training set contains 191.5K images, and both validation and testing sets include 11.0K bin-balanced images separately. The length of a bin is set to 1 year with the range from 0 to 186, and the density of each bin is from 1 to 7,149.

- **AgeDB-DIR:** AgeDB-DIR is also a imbalanced age estimation dataset proposed by Yang et al. (2021) using AgeDB dataset (Moschoglou et al., 2017). In the dataset, the age range is from 0 to 101 and the bin length is 1 year. The maximal and minimal numbers of images in a bin are 353 and 1 respectively. The training set contains 12,208 images with imbalanced densities among bins, while the validation and testing sets contain 2,140 bin-balanced images respectively.

- **NYUD2-DIR:** The NYUD2-DIR is a dataset constructed by Yang et al. (2021) for imbalanced indoor scenes depth estimation, which is an imbalanced version of NYU Depth Dataset V2 (Silberman et al., 2012). The dataset contains images of different indoor scenes and their corresponding depth maps. The bin length is set to 0.1 meters. The maximal and minimal depth is 10 meters and 0.7 meters separately in the dataset. Following Yang et al. (2021), there are 50K images in the training set and 654 images in the testing set. In each bin, the number of pixels is in the range from $1.13 \times 10^6$ to $1.46 \times 10^8$. Besides,

the number of testing pixels selected for each bin from testing images is set to 9,357 (the minimal number of pixels in each bin in the testing set), which aims to make the testing set balanced. The total number of pixels in the training set and testing set are $3.51 \times 10^9$ and $8.70 \times 10^5$ respectively.

**Baselines** We conduct experiments to compare our proposed method with the following baselines:

- **Vanilla model:** The pure backbone model that does not apply any techniques to improve the performance is presented as **VANILLA**. Following (Yang et al., 2021), we use ResNet-50 (He et al., 2016) as **VANILLA** for both IMDB-WIKI-DIR and AgeDB-DIR datasets, and ResNet-50-based encoder-decoder method (Hu et al., 2019) for NYUD2-DIR dataset.

- **Synthetic samples:** We choose existing synthetic samples methods, which are used for imbalanced regression, as baselines, including **SMOTER** (Torgo et al., 2013) and **SMOGN** (Branco et al., 2017).

- **Focal-R: FOCAL-R** (Yang et al., 2021) is the regression version of Focal loss (Lin et al., 2017), which uses a continuous function, *e.g. Sigmoid* function, to map the absolute error into the range of 0 to 1 instead of the scaling factor in Focal loss.

- **Regressor Re-Training (RRT): RRT** (Yang et al., 2021) is a two-stage training strategy inspired by Kang et al. (2020), in which the feature and classifier are decoupled.

- **Re-weighting:** We also select Re-weighting methods as baselines, including inverse-frequency weighting (**INV**) and its square-root weighting variant (**SQINV**) (Yang et al., 2021).

- **Distribution Smoothing:** Distribution Smoothing (Yang et al., 2021) includes Label Distribution Smoothing (LDS) and Feature Distribution Smoothing (FDS). Such methods are designed to utilize the information from nearby bins by smoothing the distribution of labels or features. Note that, **LDS** should be used with re-weighting methods, and we follow Yang et al. (2021) to combine **LDS** with **INV** and **SQINV**.

**Evaluation metrics** Here, we will introduce the details of all the evaluation metrics used in our experiments.

- **Mean Absolute Error (MAE):** MAE is a common evaluation metric in regression problems, which is used to measure the averaged absolute difference between ground truth and prediction results. MAE can be denoted as $\frac{1}{N} \sum_{i=1}^{N} |\hat{y}_i - y_i|$, where $N$ denotes the number of total samples, $\hat{y}_i$ and $y_i$ are the prediction result and ground truth of the $i$-th sample respectively.

- **Root Mean Squared Error (RMSE):** Similar to MAE, RMSE is also a widely used evaluation metric in regression tasks. Different from MAE, RMSE is used to measure the rooted averaged squared difference between ground truth and prediction results, which can be presented as $\frac{1}{N} \sum_{i=1}^{N} (\hat{y}_i - y_i)^2$, where $N$ denotes the number of total samples, $\hat{y}_i$ and $y_i$ are the prediction result and ground truth of the $i$-th sample respectively.

- **error Geometric Mean (GM):** GM is a new evaluation metric proposed by Yang et al. (2021). Instead of applying arithmetic mean, GM uses geometric mean over the prediction errors, which is defined as $(\prod_{i=1}^{N} e_i)^{\frac{1}{N}}$, where $e_i \triangleq |\hat{y}_i - y_i|$ denotes the $L_1$ error of the $i$-th sample.

- **Threshold Accuracy ($\delta_i$):** For NYUD2-DIR dataset, we follow Yang et al. (2021) and use standard depth estimation evaluation metric $\delta_i$ which is defined as $\max(\frac{d_i}{g_i}, \frac{g_i}{d_i}) = \delta_i < 1.25^i$, where $g_i$ and $d_i$ denote the ground truth value and the the prediction value of a pixel respectively. Here, in our experiments on NYUD2-DIR, we set $i$ to 1 as the settings in (Yang et al., 2021).

## 8.4 Implementation Details in Main Experiments

**IMDB-WIKI-DIR:** For IMDB-WIKI-DIR dataset, we use ResNet-50 (He et al., 2016) as backbone and train it with our proposed SSIR method for 90 epochs using Adam optimizer (Kingma & Ba,

2014) with $L_1$ loss. The initial learning rate is set to 1e-3 and decayed to 1e-4 and 1e-5 at the 60-th and 80-th epoch respectively. Besides, we set the batch size, $\lambda$ and $\epsilon$ to 256, 0.1 and 1e-3 separately.

**AgeDB-DIR:** For AgeDB-DIR dataset, the settings of backbone, optimizer, learning rate, loss function and batch size are the same with that of IMDB-WIKI-DIR. For the training of Focal-R+SSIR, we set the $\lambda$ and $\epsilon$ to 1.0 and 1e-3 respectively. Besides, they are set to 0.1 and 1e-3 for SQINV+SSIR.

**NYUD2-DIR:** For the experiments on NYUD2-DIR, we apply a ResNet-50-based encoder-decoder architecture (Hu et al., 2019). We train our method by Adam optimizer with 20 epochs. The initial learning rate is set to 1e-4 and decayed by 0.1 every 5 epochs. We set the batch size and weight decay to 32 and 1e-4 respectively. For $\lambda$ and $\epsilon$, we set them to 1.0 and 5e-4 for VANILLA+SSIR, and 2.5, 5e-4 for INVERSE+SSIR.

## 8.5 Implementation Details in Motivation Experiments

For the settings in the data generation of our motivation, we set the mean and variance of the two Gaussian distribution $\mu_1$, $\sigma_1$ and $\mu_2$, $\sigma_2$ to 10, 25 and -10, 25 respectively, and the dimension of the generated data $d$ is set to 100. We apply a two-layer Multi-Layer Perception (MLP) with 128 hidden neurons as the prediction model in the experiments. For the settings of the training process in the experiments, we set epochs $T_1$ and $T_2$ to 1000 and 500 respectively. The learning rates of the two training methods: *Representation Learning* and *Supervised Learning*, are set to 1e-3 and 1e-2 respectively. Besides, we set the weight decay of the training process in the experiments to 5e-5.

## 8.6 Parameter Studies

Here, we study the different parameter settings of $\epsilon$ and $\lambda$ on all three datasets and report all the results of the experiments that we conducted.

Table 4 illustrates the experiments results of different settings of $\epsilon$ and $\lambda$. The top part of the table is the results of different parameters settings of SQINV + SSIR, and the bottom part shows the results of FOCAL-R + SSIR on different settings of $\epsilon$ and $\lambda$.

The results of the parameter study of our method on AgeDB-DIR dataset are presented in Table 5. As the table shows, we present the results of different $\epsilon$ and $\lambda$ settings of our approach. The top part of the table is regarding with SQINV + SSIR, and the bottom part denotes the results of different parameter settings of FOCAL-R + SSIR.

Table 4: Parameter study of our methods on IMDB-WIKI-DIR dataset.

| $\epsilon$ | $\lambda$ | re-weighting | Loss | MAE | | | | GM | | | |
|---|---|---|---|---|---|---|---|---|---|---|---|
| | | | | All | Many | Med. | Few | All | Many | Med. | Few |
| 1e-3 | 0.01 | SQINV | $L_1$ | 7.74 | 7.15 | 12.61 | 22.94 | 4.38 | 4.13 | 7.35 | 15.04 |
| 1e-3 | 0.05 | SQINV | $L_1$ | 7.74 | 7.13 | 12.67 | 23.34 | 4.33 | 4.08 | 7.35 | 14.72 |
| 1e-3 | 0.10 | SQINV | $L_1$ | 7.63 | 7.06 | 12.17 | 23.10 | 4.26 | 4.02 | 6.82 | 14.93 |
| 1e-3 | 0.50 | SQINV | $L_1$ | 7.83 | 7.20 | 12.91 | 23.86 | 4.50 | 4.24 | 7.63 | 16.27 |
| 1e-3 | 0.10 | - | FOCAL-R | 7.88 | 7.08 | 14.75 | 25.16 | 4.39 | 4.01 | 10.20 | 18.04 |
| 1e-3 | 0.30 | - | FOCAL-R | 7.95 | 7.16 | 14.79 | 24.78 | 4.46 | 4.08 | 10.36 | 18.26 |
| 1e-3 | 0.50 | - | FOCAL-R | 8.11 | 7.25 | 15.44 | 26.76 | 4.61 | 4.20 | 11.04 | 19.46 |
| 1e-3 | 1.00 | - | FOCAL-R | 8.17 | 7.35 | 15.26 | 25.11 | 4.65 | 4.25 | 10.98 | 17.27 |

Table 6 is the parameter study regards to $\epsilon$ and $\lambda$ on NYUD2-DIR dataset. The top part of the table illustrates the results of different parameters settings of VANILLA + SSIR while the bottom part shows that of INVERSE + SSIR.

## 8.7 Training and Validation Losses

We present the training and validation loss of our proposed approach on IMDB-WIKI-DIR and AgeDB-DIR datasets. Figure 5a and 5b present the training and validation losses *w.r.t.* training epochs of SQINV + SSIR and FOCAL-R + SSIR respectively on IMDB-WIKI-DIR dataset. Figure 5c

Table 5: Parameter study of our method on AgeDB-DIR dataset.

| $\epsilon$ | $\lambda$ | re-weighting | Loss | MAE | | | | GM | | | |
|---|---|---|---|---|---|---|---|---|---|---|---|
| | | | | All | Many | Med. | Few | All | Many | Med. | Few |
| 5e-4 | 0.1 | SQINV | $L_1$ | 7.56 | 7.06 | 8.32 | 10.26 | 4.90 | 4.52 | 5.67 | 7.00 |
| 5e-4 | 0.3 | SQINV | $L_1$ | 7.67 | 6.98 | 8.86 | 10.95 | 4.88 | 4.44 | 5.84 | 7.27 |
| 5e-4 | 0.5 | SQINV | $L_1$ | 7.63 | 7.02 | 8.59 | 10.69 | 4.84 | 4.41 | 5.81 | 7.12 |
| 5e-4 | 1.0 | SQINV | $L_1$ | 7.65 | 7.01 | 8.47 | 11.49 | 4.96 | 4.58 | 5.67 | 7.22 |
| 1e-3 | 0.1 | SQINV | $L_1$ | 7.49 | 6.81 | 8.54 | 11.04 | 4.61 | 4.28 | 5.12 | 6.92 |
| 1e-3 | 0.3 | SQINV | $L_1$ | 7.73 | 7.05 | 8.74 | 11.34 | 4.86 | 4.38 | 5.97 | 7.33 |
| 1e-3 | 0.5 | SQINV | $L_1$ | 7.61 | 6.94 | 8.74 | 10.80 | 4.93 | 4.53 | 5.81 | 7.00 |
| 1e-3 | 1.0 | SQINV | $L_1$ | 7.70 | 7.02 | 8.69 | 11.37 | 4.83 | 4.37 | 5.90 | 7.34 |
| 5e-3 | 0.1 | SQINV | $L_1$ | 7.62 | 6.84 | 8.75 | 11.87 | 4.86 | 4.41 | 5.84 | 7.36 |
| 5e-3 | 0.3 | SQINV | $L_1$ | 7.65 | 6.86 | 8.83 | 11.80 | 4.88 | 4.34 | 6.05 | 8.12 |
| 5e-3 | 0.5 | SQINV | $L_1$ | 7.61 | 6.97 | 8.35 | 11.57 | 4.81 | 4.36 | 5.56 | 8.27 |
| 5e-3 | 1.0 | SQINV | $L_1$ | 8.04 | 7.15 | 9.56 | 12.20 | 5.13 | 4.50 | 6.68 | 8.37 |
| 1e-4 | 0.1 | - | FOCAL-R | 7.62 | 6.75 | 8.85 | 12.43 | 4.94 | 4.41 | 5.97 | 8.67 |
| 1e-4 | 0.3 | - | FOCAL-R | 7.38 | 6.59 | 8.26 | 12.39 | 4.79 | 4.32 | 5.46 | 8.90 |
| 1e-4 | 0.5 | - | FOCAL-R | 7.62 | 6.81 | 8.54 | 12.73 | 4.91 | 4.37 | 5.89 | 8.92 |
| 1e-4 | 1.0 | - | FOCAL-R | 7.53 | 6.66 | 8.59 | 12.73 | 4.76 | 4.14 | 5.98 | 9.46 |
| 1e-4 | 1.5 | - | FOCAL-R | 7.60 | 6.71 | 8.79 | 12.62 | 4.92 | 4.28 | 6.25 | 9.33 |
| 5e-4 | 0.1 | - | FOCAL-R | 7.69 | 6.66 | 9.30 | 13.04 | 4.90 | 4.23 | 6.26 | 9.96 |
| 5e-4 | 0.3 | - | FOCAL-R | 7.51 | 6.49 | 9.05 | 12.95 | 4.65 | 3.97 | 6.11 | 9.74 |
| 5e-4 | 0.5 | - | FOCAL-R | 7.46 | 6.59 | 8.51 | 12.83 | 4.69 | 4.13 | 5.73 | 9.13 |
| 5e-4 | 1.0 | - | FOCAL-R | 7.61 | 6.64 | 9.14 | 12.64 | 4.91 | 4.24 | 6.36 | 9.53 |
| 5e-4 | 1.5 | - | FOCAL-R | 7.37 | 6.47 | 8.74 | 12.13 | 4.72 | 4.16 | 5.87 | 8.43 |
| 1e-3 | 0.1 | - | FOCAL-R | 7.44 | 6.50 | 8.83 | 12.44 | 4.73 | 4.11 | 6.13 | 8.62 |
| 1e-3 | 0.3 | - | FOCAL-R | 7.39 | 6.45 | 8.81 | 12.37 | 4.69 | 4.11 | 5.74 | 9.26 |
| 1e-3 | 0.5 | - | FOCAL-R | 7.39 | 6.47 | 8.81 | 12.09 | 4.68 | 4.09 | 6.06 | 8.22 |
| 1e-3 | 1.0 | - | FOCAL-R | 7.35 | 6.53 | 8.51 | 11.81 | 4.62 | 4.09 | 5.64 | 8.46 |
| 1e-3 | 1.5 | - | FOCAL-R | 7.52 | 6.60 | 8.61 | 13.15 | 4.77 | 4.19 | 5.79 | 9.46 |
| 5e-3 | 0.1 | - | FOCAL-R | 7.48 | 6.54 | 8.99 | 12.10 | 4.81 | 4.21 | 5.98 | 9.06 |
| 5e-3 | 0.3 | - | FOCAL-R | 7.42 | 6.50 | 8.56 | 12.97 | 4.76 | 4.11 | 5.98 | 10.28 |
| 5e-3 | 0.5 | - | FOCAL-R | 7.60 | 6.53 | 8.98 | 13.84 | 4.89 | 4.14 | 6.40 | 11.28 |
| 5e-3 | 1.0 | - | FOCAL-R | 8.13 | 6.87 | 9.75 | 15.51 | 5.23 | 4.37 | 7.05 | 12.72 |
| 5e-3 | 1.5 | - | FOCAL-R | 9.00 | 7.33 | 11.54 | 17.79 | 5.85 | 4.68 | 8.66 | 16.16 |

Table 6: Parameter study of our method on NYUD2-DIR dataset.

| $\epsilon$ | $\lambda$ | re-weighting | Loss | RMSE | | | | $\delta_1$ | | | |
|---|---|---|---|---|---|---|---|---|---|---|---|
| | | | | All | Many | Med. | Few | All | Many | Med. | Few |
| 5e-4 | 0.1 | - | MSE | 1.542 | 0.580 | 0.871 | 2.246 | 0.669 | 0.787 | 0.739 | 0.523 |
| 5e-4 | 0.3 | - | MSE | 1.531 | 0.536 | 0.878 | 2.238 | 0.663 | 0.796 | 0.690 | 0.519 |
| 5e-4 | 0.5 | - | MSE | 1.540 | 0.566 | 0.917 | 2.239 | 0.643 | 0.778 | 0.713 | 0.481 |
| 5e-4 | 1.0 | - | MSE | 1.407 | 0.580 | 0.883 | 2.022 | 0.684 | 0.772 | 0.704 | 0.590 |
| 5e-4 | 1.5 | - | MSE | 1.475 | 0.564 | 0.872 | 2.140 | 0.692 | 0.793 | 0.723 | 0.579 |
| 5e-4 | 2.0 | - | MSE | 1.402 | 0.601 | 0.893 | 2.005 | 0.692 | 0.755 | 0.704 | 0.625 |
| 5e-4 | 2.5 | - | MSE | 1.623 | 0.535 | 0.932 | 2.380 | 0.640 | 0.801 | 0.704 | 0.455 |
| 5e-4 | 3.0 | - | MSE | 1.476 | 0.560 | 0.899 | 2.137 | 0.686 | 0.789 | 0.723 | 0.570 |
| 1e-3 | 0.1 | - | MSE | 1.476 | 0.564 | 0.919 | 2.134 | 0.693 | 0.791 | 0.723 | 0.585 |
| 1e-3 | 0.3 | - | MSE | 1.540 | 0.560 | 0.932 | 2.238 | 0.680 | 0.792 | 0.724 | 0.550 |
| 1e-3 | 0.5 | - | MSE | 1.520 | 0.568 | 0.876 | 2.212 | 0.667 | 0.787 | 0.733 | 0.521 |
| 1e-3 | 1.0 | - | MSE | 1.537 | 0.557 | 0.898 | 2.239 | 0.658 | 0.787 | 0.716 | 0.507 |
| 5e-4 | 1.0 | INVERSE | MSE | 1.365 | 0.667 | 0.930 | 1.915 | 0.685 | 0.711 | 0.717 | 0.647 |
| 5e-4 | 1.5 | INVERSE | MSE | 1.391 | 0.679 | 0.849 | 1.968 | 0.670 | 0.713 | 0.734 | 0.602 |
| 5e-4 | 2.0 | INVERSE | MSE | 1.313 | 0.749 | 0.919 | 1.795 | 0.700 | 0.702 | 0.742 | 0.682 |
| 5e-4 | 2.5 | INVERSE | MSE | 1.291 | 0.766 | 0.894 | 1.756 | 0.705 | 0.669 | 0.763 | 0.718 |

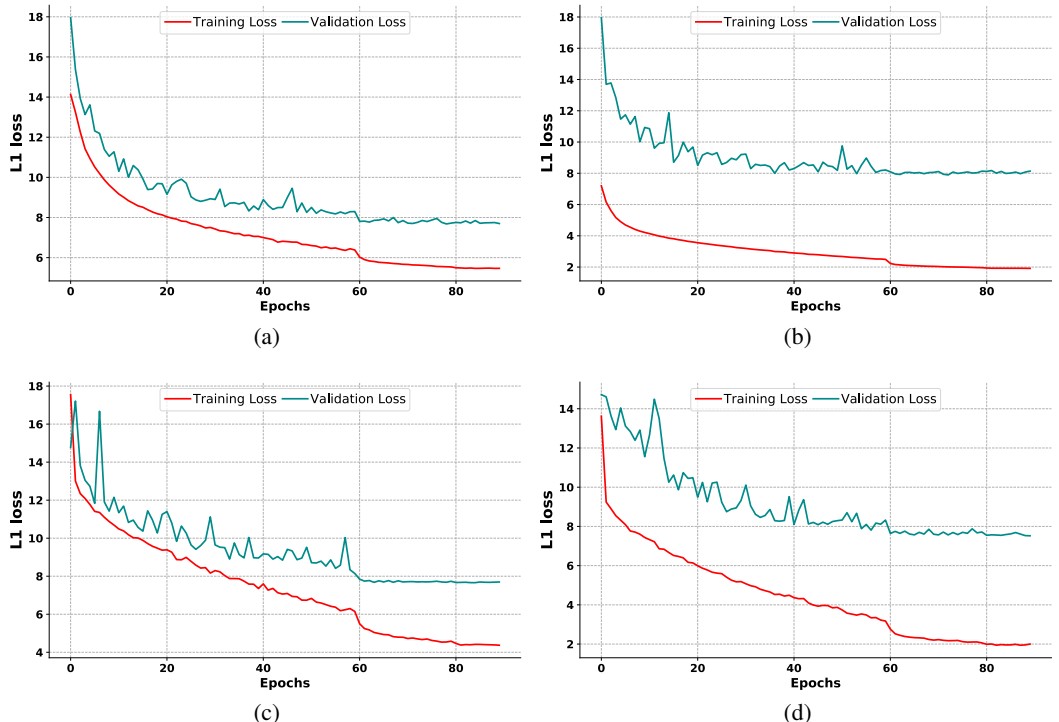

Figure 5: (a) Training and validation loss *w.r.t* training epochs of SQINV + SSIR on IMDB-WIKI-DIR dataset. (b) Training and validation loss *w.r.t* training epochs of FOCAL-R + SSIR on IMDB-WIKI-DIR dataset. (c) Training and validation loss *w.r.t* training epochs of SQINV + SSIR on AgeDB-DIR dataset. (d) Training and validation loss *w.r.t* training epochs of FOCAL-R + SSIR on AgeDB-DIR dataset.

and 5d illustrate the training and validation losses of SQINV + SSIR and FOCAL-R + SSIR on AgeDB-DIR dataset.

## 8.8 FEATURE BALANCE UNDER REGRESSION SENSE

To qualitatively and quantitatively evaluate the balance of features under the regression sense, we first attempt to offer a definition of feature balance, and then assess the impact of our approach and the baseline approaches experimentally.

### 8.8.1 DEFINITION OF FEATURE BALANCE

As mentioned in (Kang et al., 2021), the feature space can be more balanced if the features from the same class or bin are more similar, and the balanced feature space can handle the imbalanced problem well. In this subsection, we give the definition of feature balance under regression sense. We assume that $z$ with size $c(z)$ is a feature vector extracted from the neural network corresponding to the input $x$. It is often the case that $z$ is semantically rich. Each element in $z$ signifies a different attribute. Without loss of generality, it is also reasonable to expect that the elements in $z$ are mutually independent. And then, we assume that for data within a bin, its features all conform to a Gaussian distribution $\mathcal{N}(\mu_i, \sigma_i^2), \forall i = 1, 2, ..., c(z)$ for each dimension. The reasons for the Gaussian assumption are: Gaussian distribution is the most commonly used, and it can fit deep learning features relatively well (Wang et al., 2017; Sun et al., 2020; Louizos & Welling, 2016). Therefore, the variances of the distributions of each dimension can be used as the evaluation metric, to evaluate the imbalanced ratio

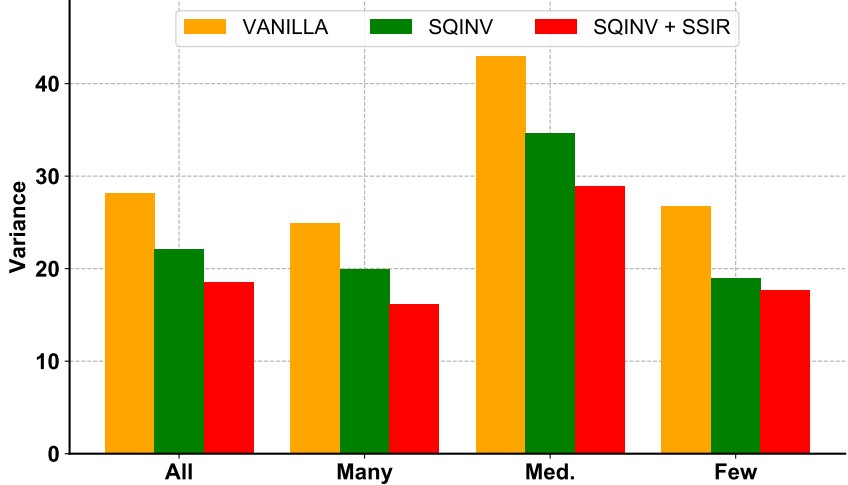

Figure 6: Visualization of variance values of deep learning features on the IMDB-WIKI-DIR testing dataset for the VANILLA, SQINV and SQINV+SSIR methods. We compare the three methods on all bins, many-shot bins, middle-shot bins and few-shot bins respectively. SQINV+SSIR owns a smaller variance $\sigma$, which implies more balanced features are leaned.

of the feature space:

$$\sigma = \sum \sigma_i \tag{14}$$

$$s.t. \quad i = 1, 2, ..., c(\boldsymbol{z}) \tag{15}$$

where smaller $\sigma$ indicates more balanced feature space that can handle the imbalanced problem well.

### 8.8.2 EVALUATION OF FEATURE BALANCE ON IMDB-WIKI-DIR DATASET

We conduct experiments on IMDB-WIKI-DIR testing dataset, recall that previously we used $f(\boldsymbol{x}; \boldsymbol{\theta})$ to denote a neural network. Here $f(\boldsymbol{x}; \boldsymbol{\theta})$ is implemented via ResNet-50. $\boldsymbol{z}$ is the feature extracted from the penultimate layer in $f(\boldsymbol{x}; \boldsymbol{\theta})$, which has a dimension of 2048. The methods we use for evaluation are **VANILLA**, **SQINV** and our **SQINV+SSIR**. In Figure 6, we show the variance values for the different methods on many-shot, middle-shot and few-shot classes, it is evident that our method yields a much smaller variance $\sigma$, which means the feature space generated by our method is more balanced. Figure 7 shows more variance values $\sigma$ of the three methods within different bins.

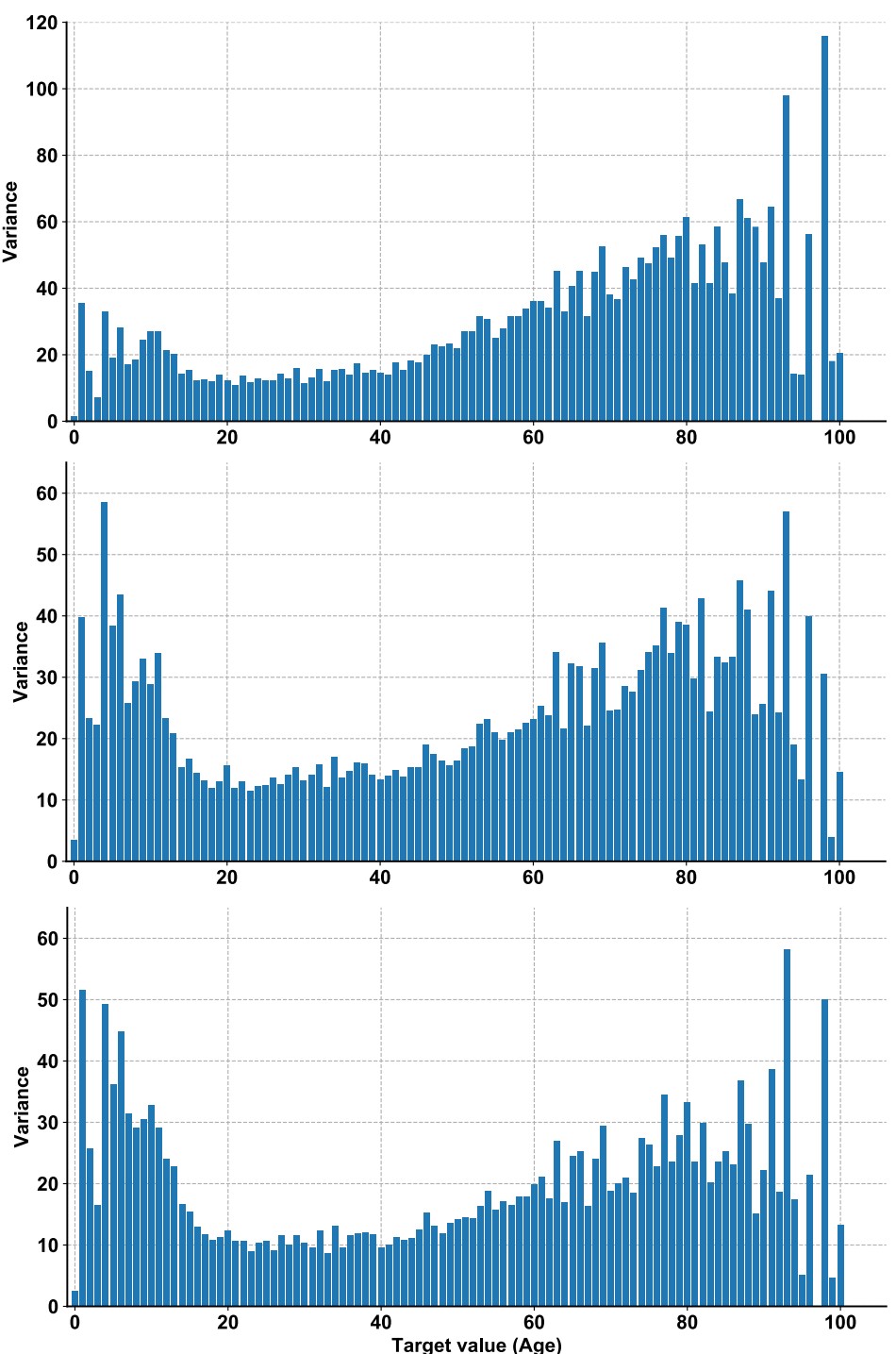

Figure 7: Visualization of variance values $\sigma$ of deep learning features on the IMDB-WIKI-DIR testing dataset for the VANILLA, SQINV and SQINV+SSIR methods on different bins. The three subfigures from the top to bottom correspond in turn to VANILLA, SQINV and SQINV+SSIR methods.

