# OpenReview forum: "Lifting Imbalanced Regression with Self-Supervised Learning"
_ICLR.cc/2022/Conference — ICLR 2022 Submitted_

### Official Review · Reviewer_hPS1 · 2021-10-20

**Correctness:** 3
**Technical Novelty And Significance:** 3
**Empirical Novelty And Significance:** 4
**Recommendation:** 6
**Confidence:** 5

**Main Review:**

Positive points:
1. This paper studies an important and practical task, namely long-tailed regression. The importance of this task can also be supported by a recent survey of deep long-tailed learning [1], which highlights long-tailed regression as an important future direction.


2. This work explores contrastive self-supervised learning (CSL) for imbalanced regression, and finds the difference between CSL used for classification and CSL used for regression. That is, the regression is more sensitive to data augmentation (i.e., adding noise).  This may inspire future research when using CSL for imbalanced regression.


3. This paper broadens the concept of similarity and dissimilarity between noise samples and original samples from classification to regression, and develops a new self-supervised learning based method for long-tailed regression. Extensive results verify the effectiveness of the proposed method.


Negative points:
1. In the definition of similarity and dissimilarity, it is unclear whether the model should be well-trained when evaluating the similarity or dissimilarity of two samples. Moreover, what is the distance function D? Do different distances influence a lot? The definition of the similarity and dissimilarity in the regression would be more solid if the authors can make these concepts more clear.


2. It is unclear why the proposed self-supervised regression helps to handle imbalance. Based on the current method, it seems that self-supervised regression helps to train a more noise-invariant model. Although such an operation may empirically improve the performance of long-tailed regression (on some baselines in experiments), it is unclear how it helps to handle class imbalance. Does it learn a more balanced feature space? I am okay if the authors want to leave this as future work, but I still expect the authors to discuss it.


3. Since this study explored a self-supervised learning based method for long-tailed regression and discussed contrastive self-supervised learning, I really suggest the authors to review more recent contrastive learning based long-tailed studies, in addition to the mentioned KCL (Kang 2021). According to the survey [1], contrastive learning based long-tailed methods also include Hybrid [2], PaCo[3] and DRO-LT [4].



Minor suggestions:
1. In the abstract, the noise is not defined and may confuse readers. The writing can be further improved.


2. The caption of figure 1 is a little confused, and I suggest the authors further improve it.


3. In the first paragraph of Section 5, it would be better to add citations for baselines.


References:

[1] Deep long-tailed learning: A survey. ArXiv, 2021.

[2] Contrastive learning based hybrid networks for long-tailed image classification. In CVPR, 2021.

[3] Parametric contrastive learning. In ICCV, 2021.

[4] Distributional robustness loss for long-tail learning. In ICCV, 2021.



**Summary Of The Paper:**

This paper studies long-tailed regression and explores self-supervised learning to alleviate class imbalance. More specifically, the work broadens the concept of similarity and dissimilarity between noise samples and original samples from classification to regression, and develops a new self-supervised learning based method for long-tailed regression.

**Summary Of The Review:**

Overall, I like this paper because of its task importance and the new formulation of self-supervised learning for long-tailed regression. If the authors can fix the negative points mentioned above, this paper would be more solid and thorough.

---

> ### Author Response · Authors · 2021-11-10
> **Response to reviewer hPS1**
>
> We thank the reviewer for your constructive comments. We are encouraged that you like this paper because of its task importance and the new formulation of self-supervised learning for long-tailed regression. We answer your questions below, and we will incorporate all feedback in revision.
>
> **Q1: In the definition of similarity and dissimilarity, what is the distance function D? Do different distances influence a lot? The definition of the similarity and dissimilarity in the regression would be more solid if the authors can make these concepts more clear.**
>
> We first consider the most simple but common distance function in regression tasks, such as l1 and l2 distance functions. In our experiments, we found that different distance functions had little effect on final performance. For the selection of l1 or l2 in different datasets, we follow the settings in [1]. We will give more details about the definition of similarity and dissimilarity in the revision.
>
> **Q2: Does it learn a more balanced feature space?**
>
> Our inspiration for using SSL comes from the positive impact of the balanced feature space learned by SSL in the imbalanced classification problem. Therefore, we explored whether SSL has a similar positive effect in regression tasks.  Specifically, we first made a toy experiment in motivation to verify this hypothesis. The results of this experiment show that SSL can effectively alleviate the negative impact of imbalanced problems. Moreover, we obtained similar results in real-world data sets. As for whether balanced feature space is working, we will further explore it in our future research.
>
> **Q3: Review more recent contrastive learning based long-tailed studies.**
>
> We have included and discussed these in our related work in this revision.

---

> > ### Comment · Reviewer_hPS1 · 2021-11-19
> > **Two remaining questions**
> >
> > Thanks for the response, which has addressed most of my concerns. There are still two questions remaining.
> >
> > 1. In the definition of similarity and dissimilarity, it is unclear whether the model should be well-trained when evaluating the similarity or dissimilarity of two samples. This may influence the model training during the early stage.
> >
> > 2. It is unclear why the proposed self-supervised regression helps to handle imbalance. Can you provide more analysis regarding this, e.g., any visualization?

---

> > > ### Author Response · Authors · 2021-11-20
> > > **Answers to the two remaining questions**
> > >
> > > **Q1: it is unclear whether the model should be well-trained.**
> > >
> > > The model does not need to be well-trained when evaluating the similarity or dissimilarity of two samples, as stated in our definition.
> > >
> > > **Q2: Can you provide more analysis regarding why the proposed self-supervised regression helps to handle imbalance?**
> > >
> > > Thank you for your suggestion. We have provided some analysis of the balance of features in our latest revision. In order to analyze the effect of different methods on the balance of features, we first give the definition of feature balance in the regression sense. Afterward, we assume that the features follow a Gaussian distribution, and we use variance as a measure of feature balance. We calculate the corresponding variance within each bin on many-shot, middle-show and few-shot classes. Finally, the experimental results on the IMDB-WIKI-DIR testing dataset showed that SSL can indeed learn more balanced features. We present more details in the section 8.8 in appendix.

---

> > > > ### Comment · Reviewer_hPS1 · 2021-11-25
> > > > **Response**
> > > >
> > > > Thanks for further clarification.

---

### Official Review · Reviewer_cjZs · 2021-11-02

**Correctness:** 3
**Technical Novelty And Significance:** 3
**Empirical Novelty And Significance:** 3
**Recommendation:** 6
**Confidence:** 3

**Main Review:**

Strength:

1. Although the self-supervision signal has been discussed under the context of classification, leveraging it in the imbalance regression task is novel.
2.  The paper identifies and addresses an important problem: how to define positive pairs for the regression task and what is a valid augmentation to produce a positive pair. Solving the problem is a necessary step to introduce more self-supervised methods into the imbalanced regression domain.
3. The solution proposed in the paper is novel. Augmenting on the label space and reversely finding the mapping in the image space is interesting. The method also shows promising empirical results in Tables 1,2, 3.

Weakness:

1. It is unclear why only noise-based augmentation is discussed in the paper.

   The paper only discusses noise-based augmentation. However, random noise is only one of the many augmentation techniques. There are a number of other augmentations popular in self-supervised contrastive learning, for example, random cropping, random blur, and random color distortions. Can they transfer to the regression task? Can the proposed method apply to these augmentation techniques as well?
2. The argument about "infeasible migration" in the Introduction is weakly supported and lacks empirical evidence.

   The paper argues that "In the case of regression, however, it's not at all feasible when a similar migration was made (from classification to regression) ". The authors support the argument using an illustration in Figure 1, where regression models make drastically varying predictions to different noises and hence prove that the random noises ambiguate regression targets.

   However, the support is a little weak considering that a classifier will also make wrong predictions when noises are added [1]. For example, in Figure 1 (a), the cat could be classified as a dog/bird when strong enough noise is added. In another word, the illustration does not fully convince me why the use of noise should be different on classification and regression and why the "migration" is not feasible. Why would simple thresholding on the noise level like in classification [2] fail on the regression task?

   Moreover, the empirical evidence for the argument is missing in the paper. There lacks a contrastive learning baseline that uses random noise to produce positive pairs like in the classification tasks. Compared with the sophisticated data generation procedure proposed in the paper, it would be interesting to know how a naive random noise augmentation performs.
3. The settings in Motivation (Section 3) and Method (Section 4) are inconsistent, which undermines the paper's motivation.

   In the motivation section, self-supervision is used as a pre-train and steady improvement is observed in Figure 2. However, in the method section, self-supervision is used as an additional training objective (Equation 11). Why not keep the pre-training setting in the method?

4. There are some minor issues with the parameter study.

   The paper concluded that the proposed method achieves the best performance with $\epsilon = 1e-3, \lambda=1.0$. However, no results are provided when $\lambda>1$.



[1]: Zheng et al., Improving the Robustness of Deep Neural Networks via Stability Training, CVPR 2016

[2]: Chen et al., A Simple Framework for Contrastive Learning of Visual Representations, ICML 2020.




**Summary Of The Paper:**

This paper proposes to leverage self-supervision to improve imbalanced regression. An augmentation method is designed to keep the regression target after augmentation close to the original target.

**Summary Of The Review:**

I recommend a borderline acceptance. Although the motivation of the paper is not fully justified and some discussions are too limited to a specific domain (random noise augmentation), the paper explores a new topic and provides a technically sound solution. My overall attitude towards the paper is positive.

---

> ### Author Response · Authors · 2021-11-10
> **Response to reviewer cjZs**
>
> We thank the reviewer for your thoughtful feedback. We are pleased to see that you found that we explored a new topic and provided a technically sound solution. We address your concerns below, and we will incorporate all feedback in the revision.
>
> **Q1: Can other augmentation methods transfer to the regression task?**
>
> Noise-based augmentation is a common and straightforward augmentation method, so we first consider this augmentation method. Other augmentation methods, such as random cropping, random blur, and random color distortions, also can be applied in our methods. We are exploring the automated noise learning method, *i.e.*, learn to augment. In the future, we will try to use other data augmentation methods in imbalanced regression.
>
> **Q2: Why the use of noise should be different on classification and regression? Why would simple thresholding on the noise level fail on the regression task?**
>
> 1. Due to the boundary between classes being obvious, and the outputs of classification models are probability values of each class, after adding noises, the change of probability values of each class may not affect their ranking. Therefore, the classification results may not be affected. However, in regression tasks, the regression values may deviate greatly after adding noise to samples.
>
> 2. It is ok to set a simple threshold on the noise level. However, in this case, it is difficult to determine a specific threshold to guarantee the noisy outputs are meaningful (the distance between original outputs and augmented outputs is limited within the threshold according to the definition of similarity and dissimilarity), due to the regression model is more sensitive to noise as we mentioned above. If we want to determine a specific threshold based on which the noisy outputs are meaningful, we need to compute the distance between the original outputs and noisy outputs to adjust the threshold accordingly. So why don't we directly add noises, which have been limited within the threshold, to original outputs and use such noisy outputs to generate noisy inputs?
>
> **Q3: There lacks a contrastive learning baseline. It would be interesting to know how a naive random noise augmentation performs.**
>
> In fact, our work was inspired by recent advances in contrastive learning on classification tasks. Note, however, that the way we have combined SSL and imbalance regression is innovative. Specifically, we add noise to the output values under the definition of similarity in regression, and later obtain noisy samples by back-propagation. Rather than directly adding random noise to the input samples.
>
> To compare with naive random noise-based contrastive learning method, we conducted a comparison experiment on the AgeDB-DIR dataset. The setup of the comparison experiment is following: we add random noise directly to the input samples and provide results combined with FOCAL-R and SQINV; except for this, we keep the other hyperparameters consistent with our experiments.
>
> With the addition of random noise, the combined SSIR with FOCAL-R and SQINV approaches can achieve the performance on MAE with **7.57** and **7.71** respectively. In contrast, the combined SSIR with FOCAL-R and SQINV approaches with noise generation proposed in our paper can achieve MAE performance with **7.35** and **7.49** respectively. The experimental results demonstrate the superiority of our noise generation approach.
>
> **Q4: The settings in Motivation (Section 3) and Method (Section 4) are inconsistent.**
>
> In our main experiments, we first conducted experiments with pre-trained SSL, and then, we found that using SSL as an additional training objective could achieve similar or even better results. In addition, employing it as an additional training objective can take less training time due to pre-trained SSL being a two-stage training approach.
>
> **Q5: There are some minor issues with the parameter study.**
>
> Thank you for your suggestion, we have provided some results with $\lambda >1$ in the parameter study of the revision. Further details can be found in subsection 8.6 in the appendix of our updated revision, we have also included a comparison of this case in Figure 3a.

---

### Official Review · Reviewer_QHt8 · 2021-11-02

**Correctness:** 4
**Technical Novelty And Significance:** 3
**Empirical Novelty And Significance:** 3
**Recommendation:** 5
**Confidence:** 3

**Main Review:**

Strengths:
This paper seamlessly combines self-supervised learning and imbalanced regression by giving the formal definition of similarity in the regression task, and proves that self-supervised learning really relieve the long-tailed regression problem. The Experiments are substantial.

Weaknesses:
1)  In the penultimate paragraph of Section 4: “On the other hand, considering that a single back-propagation optimization may not be an accurate approximation, we therefore performed a two-step optimization of r.” Why a two-step optimization can be helpful for obtaining an accurate approximation? Please give an explanation.
2)  In Section 4, why the authors apply noise to the output instead of the input? How to solve the imbalanced regression problem when applying noise to the input?
3)  In Section 3, Data generation, why the test set generated by the authors is balanced? How would the proposed method perform if the test data is unbalanced?
4)  In Section 3, the fourth line of second paragraph: y=w^Tx, the vector transpose symbol ‘T’ should not be bolded.
5)  The sentence: “the left image suggest a greater change with augmented data.” “suggest” should be “suggests”.



**Summary Of The Paper:**

This paper proposes a novel algorithm SSIR to address the imbalanced problem in regression tasks, which seamlessly combines self-supervised learning and imbalanced regression by giving the formal definition of similarity in the regression task. Besides, the authors specifically propose to limit the volume of noise on the output, and in doing so to find meaningful noise on the input by back propagation. Experimental results show that our approach achieves the state-of-the-art performance.

**Summary Of The Review:**

I think this paper in its present form cannot be accepted for publication in ICLR.

---

> ### Author Response · Authors · 2021-11-10
> **Response to reviewer QHt8**
>
> We thank the reviewer for your insightful and positive feedback. We are encouraged that you found our method novel and experiments substantial. Meanwhile, we are glad you discovered that we are the first to seamlessly combine SSL and imbalanced regression by giving the formal definition of similarity in the regression task, and proving that SSL really relieves the long-tailed regression problem. We address your comments below and will incorporate all feedback.
>
> **Q1: Why a two-step optimization can be helpful for obtaining an accurate approximation.**
>
> To quantitatively compare the benefits of two-step approximation against one-step approximation, we conducted a comparison experiment on the AgeDB-DIR dataset. The setup of the comparison experiment is the following: we perform a one-step optimization of **r** and provide results combined with SQINV; except for this, we keep the other hyperparameters consistent with our experiments.
>
> When performing one-step optimization,  the combined SSIR with SQINV approach can achieve the performance on MAE with **7.62**. In contrast, the combined SSIR with SQINV approach with a two-step approximation can achieve MAE performance with **7.49**. The experimental results demonstrate the superiority of two-step approximation.
>
> However, we have found in our experiments that there is no significant increase in performance but an increase in computational overhead when the number of optimization steps is greater than or equal to 3. Therefore, in order to balance accuracy and efficiency, we have used two-step optimization in our paper.
>
> **Q2: why the authors apply noise to the output instead of the input?**
>
> The definitions of similarity and dissimilarity between original and augmented samples in regression have not been clearly defined. This is one of the problems solved in our paper, and also one of our contributions. After we give the definition, there will be a problem in migrating SSL from classification to regression. That is, the premise of SSL is to have similar samples, but we find that after adding random noises to the input samples, it is not guaranteed that we can get similar samples. Therefore, we propose to directly add noises to the outputs of the model, and get the meaningful noisy samples by back propagation, which has been described in the third paragraph of the Introduction section, and also illustrated pictorially in Figure 1(b).
>
> **Q3: How to solve the imbalanced regression problem?**
>
> There have been several methods proposed in [1] to solve this problem, such as Focal-R, SQINV, etc. These methods try to solve this problem by reweighting the loss function or resampling few-shot classes. Our main contributions are:
>
> + To the best of our knowledge, we are the first to broaden the concept of similarity and dissimilarity between augmented samples and original samples from classification to regression. We are also the first to reveal imbalanced regression problems with SSL.
> + Having been convinced of the beneficial effect of applying SSL on a simple experiment, we extend it to the neural network-based long-tailed regression task. To ensure that the noise is manageable with respect to the regression's similarity, we propose generating noise on regression values within a predefined threshold.
> + We obtain an optimization problem on noise with an efficient approximate solution algorithm based on a first-order Taylor expansion. The augmented input samples are obtained by back-propagation.
> + As a practical trial for the proposed method, the results reveal that we attained not only the best results, but also the possibility to mix it freely with other training techniques.
>
> **Q4:  why the test set generated by the authors is balanced?**
>
> This task was proposed by Yang et al. [1]. In the settings of their paper, the training set is imbalanced while validation and testing sets are balanced, which has been elaborated in **problem statement** in our paper. Actually,  this is a also common setting in the context of imbalanced classification, and there has been lots of work trying to solve the imbalanced recognition under this setting.
>
> **Q5: the vector transpose symbol ‘T’ should not be bolded.**
>
> Thank you very much for your suggestions, we have fixed these typos in the revision.
>
> **Q6: The sentence: “the left image suggest a greater change with augmented data.” “suggest” should be “suggests”**
>
> Thank you very much for your suggestions, we have fixed these typos in the revision.
>
> [1] Yuzhe Yang et al. Delving into deep imbalanced regression. ICML 2021.

---

### Official Review · Reviewer_TyLX · 2021-11-03

**Correctness:** 3
**Technical Novelty And Significance:** 3
**Empirical Novelty And Significance:** 2
**Recommendation:** 5
**Confidence:** 4

**Main Review:**

Pros:

[1] The imbalanced regression problem studied by this paper is of great significance not only to the academic research but also to the real-world applications.

[2] Authors did comprehensive experiments on several popular benchmarks to show the effectiveness of the paper.

[3] How to do noise generation, which ensures a high degree of similarity between the disturbed sample and the original sample, is very intuitive and easy to follow.

Cons:

[1] The performance improvements to current state-of-the-art method is quite marginal. For NYUD2-DIR dataset, the improvements to RMSE and delta1 are 0.05 and 0.0, respectively. For AgeDB-DIR dataset, the improvements to MAE and EM are 0.12 and 0.16, respectively. For IMDB-WIKI-DIR dataset, the improvements to MAE and GM are 0.15 and 0.11, respectively.

[2] The performance of the proposed method on few-shot classes is much worse than the previous SOTA on IMDB-WIKI-DIR dataset, as shown in Table 1. Is there any reason to explain it?

[3] In Figure 1(a) and Figure 2(b), the author drew several conclusions about noise-related issues, including the sensitivity to noise and the impact of noise, based only on one selected sample. Such a special example is difficult to be convincing enough, especially when we are dealing with regression tasks that contain many outlier/noisy samples. My suggestion is that the author can try to provide statistical analysis on the overall datasets, and quantitatively compare the benefits of noise generation method against vanilla random noise method.

[4] The idea of integrating self-supervised learning with imbalanced recognition has been investigated by previous works (1). Although this paper is working on a different task, i.e. regression, the technical contribution of how to integrate self-supervised learning with imbalanced learning is relatively limited.

[5] There are several typos in the paper, for example, "Update ri by by taking" in the algorithm workflow "Algorithm 1: Self-Supervised Imbalanced Regression (SSIR)" should be "Update ri by taking". Authors may want to double check the draft.

(1) Rethinking the Value of Labels for Improving Class-Imbalanced Learning. NuerIPs 2020.

**Summary Of The Paper:**

This paper researched on the recently proposed long-tailed regression problem with self-supervised learning method. Two questions are investigated by this paper: 1) how to measure the similarity and dissimilarity under the regression sense; 2) it is not guaranteed that the sampled with perturbations are similar to the original samples without any noise. The former problem is addressed by providing a formal definition of similarity and the later question is addressed by limiting the volume of noise on the output. Authors report the results on three datasets to demonstrate the superiority of our proposed methods, including IMDN-WIKI-DIR, AgeDB-DIR and NYUD2-DIR.

**Summary Of The Review:**

Although the imbalanced regression problem is very interesting and import, I have concerns on the technical contribution and effectiveness of the method. Therefore, my current rating is: "5: marginally below the acceptance threshold"

---

> ### Author Response · Authors · 2021-11-10
> **Response to reviewer TyLX**
>
> Many thanks to the reviewer for your constructive feedback. We are glad that you found our approach very intuitive and easy to follow, and evaluated with adequate experiments. We are encouraged that you agree that the problem of imbalanced regression is very important for both practical applications and academic research. We address your comments below and will incorporate all feedback.
>
>
> **Q1: The performance improvements to current state-of-the-art method is quite marginal.**
>
> Actually, we consider that the improvement is significant. We are mainly comparing the first paper [1] in this field following the context of neural networks. In their experiments, on AgeDB-DIR dataset, **when compared with VANILLA**, their methods improve 0.3 and 0.34 on MAE and GM respectively. On NYUD2-DIR dataset, **compared with VANILLA**, their methods improve 0.139 and 0.028 on RMSE and $\delta_1$ respectively. On IMDB-WIKI-DIR dataset, **compared with VANILLA**, their methods improve 0.41 and 0.26 on MAE and GM. And In our experiments, **please note that the performance of our method improves again compared with the SOTA methods** in [1]. So the performance of our method improves significantly. Besides, such improvements are evaluated by different evaluation metrics, which show the effectiveness of our method in different aspects.
>
> **Q2: Provide statistical analysis on the overall datasets, and quantitatively compare the benefits of noise generation method against vanilla random noise method.**
>
> To quantitatively compare the benefits of our noise generation method against vanilla random noise method, we conducted a comparison experiment on the AgeDB-DIR dataset. The setup of the comparison experiment is following: we add random noise directly to the input samples and provide results combined with FOCAL-R and SQINV; except for this, we keep the other hyperparameters consistent with our experiments.
>
> With the addition of random noise, the combined SSIR with FOCAL-R and SQINV approaches can achieve the performance on MAE with **7.57** and **7.71** respectively. In contrast, the combined SSIR with FOCAL-R and SQINV approaches with noise generation proposed in our paper can achieve MAE performance with **7.35** and **7.49** respectively. The experimental results demonstrate the superiority of our noise generation approach.
>
> **Q3: The idea of integrating SSL with imbalanced recognition has been investigated by previous works.**
>
> Though SSL has been used to solve the problem of imbalanced classification, there are still significant differences between previous works with our method. Such differences can be summarized as follows:
>
> 1. As described in our paper, there is no obvious boundary between regression values in regression tasks. However, there are great differences between different classes in classification tasks. Therefore, regression tasks are more sensitive to noise than classification tasks, as shown in Figure 1 (a) in our paper. Due to the obvious boundary between classes, the outputs of the classification model are probability values of each class, after adding noise, the change of probability values of each class may not influence the classification results. However, in regression tasks, the regression values may deviate considerably after adding noise to samples.
>
> 2. The definitions of similarity and dissimilarity between original and augmented samples in regression have not been clearly defined. This is one of the problems solved in our paper, and also one of our contributions. After we give the definition, there will be a problem in migrating SSL from classification to regression. That is, the premise of SSL is to have similar samples, but we find that after adding random noises to the input samples, it is not guaranteed that we can get similar samples. Therefore, we propose to directly add noises to the model's outputs and get the meaningful noisy samples by back propagation, which has been described in the third paragraph of the Introduction section and illustrated pictorially in Figure 1(b).
>
> **Q4: The performance of the proposed method on few-shot classes is much worse than the previous SOTA on IMDB-WIKI-DIR dataset.**
>
> In Table 1, the combinations of our method with FOCAL-R and SQINV can both achieve the best performance. Compared with FOCAL-R-based methods, the combination of FOCAL-R and our method can achieve the best performance in few-shot classes. Compared with other SQINV methods, the combination of SQINV and our method performs worse than SOTA. This is maybe caused by the bias of the dataset. In fact, both our experiments and the previous experiments in [1] show that no loss function can perform well on many-shot, middle-shot, and few-shot tasks at the same time.
>
> **Q5: There are several typos in the paper.**
>
> Thank you very much for your suggestions, we have fixed these typos in the revision.
>
> [1] Yuzhe Yang et al. Delving into deep imbalanced regression. ICML 2021.

---

### Author Response · Authors · 2021-11-10
**Summary of the updates in the revision**

We thank all reviewers for their constructive comments. During the rebuttal period, we revised the paper to faithfully reflect the comments from all the reviewers, performing multiple sets of experiments. We have made the following updates to the revision.
1. We have reorganized and polished the abstract and introduction sections. We hope that our paper will give a clearer picture of our motivations and methods, even to readers who are not very familiar with the field.
2. We add two experiments in the appendix: (1) the experiment of directly adding random noise on the inputs (2) the experiment of the one-step optimization method.
3. We emphasize our contributions in the introduction section, and we summarize the contributions of our paper into four aspects.
4. We revise the definitions of similarity and dissimilarity in regression to make the definition more strict and clear.
5. We redraw the figures in Figure 1 to make them look more clear. In addition, the caption in Figure 1 has also been revised to better describe the figures, hoping to better reflect the contents of the pictures.
6. We fix some typos in our paper and some grammatical errors.
7. We include and discuss some related work.

---

### Author Response · Authors · 2021-11-17
**More Questions and Discussions are Welcomed**

Dear reviewers:

Thanks a lot for your efforts in reviewing this paper. We tried our best to address all mentioned concerns/problems, and have uploaded a new version of our paper incorporating all suggestions raised by four reviewers to improve our paper. Are there any unclear explanations here? We are happy to further clarify them.

Best,

Authors

---

### Author Response · Authors · 2021-11-22
**Some Improvements to Our Paper**

We thank all reviewers for the positive feedback and helpful comments. We have revised our paper by taking into account their suggestions. In addition, we have tried our best to resolve your concerns. These major concerns are listed below:
+ Effectiveness of the method: We have listed the improvements of the previous methods, and in comparison with the previous best method, the improvement of our method is significant.
+ Technical contribution: We illustrate the difficulties in migrating SSL from classification to regression, and we are the first to address these problems comprehensively.
+ Comparison of the variants, random noise and one-step optimization: Experimental results show the effectiveness of the two-step optimization, and it also illustrates the strategy of adding noise to the output to obtain augmented samples is quite valid.
+ Parameter study: We investigate the impact of different hyperparameters in the appendix and give detailed experimental results.
+ Discussion of feature balance: We give a measure of feature balance in the context of regression, experimental results demonstrate that our method can indeed improve the balance of features.
+ More discussions on related works: We have carefully discussed the linkages and differences between the related works.
+ typos: We have fixed these typos in our latest revisions.
+ Empirical evidence to support some of the arguments: In addition to confirming our arguments experimentally, we discuss these more thoroughly.

We hope that our answers have helped you better address your concerns. Please feel free to ask us if you have further questions, and we will be happy to answer them.

---

### Author Response · Authors · 2021-11-26
**Any further concerns or suggestions?**

Thank you so much for the great time and effort, we have taken your comments into account and tried our best to address your concerns. Our work is more solid and thorough now.

If there are any further concerns, please feel free to raise them. We will be delighted to address these concerns for you.

---

### Decision · Program_Chairs · 2022-01-20

**Decision:**

Reject

**Comment:**

This paper proposes to use self-supervised learning in the context of "imbalanced regression", where some values of the outcome variables are rare, such as in long-tailed regression. The author's proposal can be interpreted as a Monte Carlo approximation of a density smoothing technique, akin to Yang et al. 2021. They test their approach on three datasets. Overall, it provides marginal improvements, whose statistical significance are not assessed. All reviewers agreed that the paper has merits but that it should be further improved to demonstrate that the proposed method is indeed a step forward  in solving the problem of imbalanced regression. The authors should also provide stronger motivation for their pipeline details and experimental setup choices. I therefore recommend rejection, with encouragement for improvement in two directions: strengthening the experimental section, in particular by assessing statistical significance, and by improving the writing of the paper by developing a more rigorous exposition.